# Learning and cognition in highspeed decision making

Martin Krause, Wolfram Schulze, Stefan Schuster*

Department of Animal Physiology, University of Bayreuth, Bayreuth, Germany

## eLife assessment

This **important** study investigates the adaptability of prey capture by archerfish, which hunt insects by spitting at them and then rapidly turning to reach their landing point on the water surface. The results of elaborate behavioral experiments and measurements show that, even though the visuomotor behavior unfolds very rapidly (in less than 100 ms), it is not hardwired and can adapt to different simulated physics and different prey shapes. The data are **convincing** and should be of relevance to those interested in rapid decision making in general, beyond the archerfish model.

*For correspondence:
stefan.schuster@uni-bayreuth.de

Competing interest: The authors declare that no competing interests exist.

**Abstract** It is widely accepted that more time and information yield better decisions. However, some decisions manage to be extremely fast and yet accurate. The trick of such highspeed decisions appears to be the use of simplifying heuristics that works well for the most common condition but lacks flexibility otherwise. Here, we describe an unexpected level of flexibility in a complex highspeed decision that is made faster than an Olympic sprinter can respond to the start gun. In this decision, archerfish observe the initial speed, direction, and height of falling prey and then use these initial values to turn right towards where ballistically falling prey would later land. To analyze the limits in flexibility of this highspeed decision, we developed and critically tested a system that allowed us to replace the usual ballistic relation between initial prey motion and the expected landing point with another deterministic rule. We discovered that, surprisingly, adult fish could reprogram their highspeed decision to the new rule. Moreover, after reprogramming their decision fish were immediately able to generalize their decision to novel untrained settings, showing a remarkable degree of abstraction in how the decision circuit represented the novel rule. The decision circuit is even capable of simultaneously using two distinct sets of rules, one for each of two visually distinct objects. The flexibility and level of cognition are unexpected for a decision that lacks a speed-accuracy tradeoff and is made in less than 100 ms. Our findings demonstrate the enormous potential highspeed decision making can have and strongly suggest that we presently underappreciate this form of decision making.

## Introduction

It is common wisdom that the quality of our decisions depends on how much time and effort we devote to them. The expected trade-off between the speed of a decision and its accuracy is very well documented not only in humans but for many decisions across many species from slime mold to primates (*Chittka et al., 2009*). A class of very rapid decisions appears, however, to deviate from this rule (e.g. *VanRullen and Thorpe, 2001*; *Romo and Salinas, 2001*; *Uchida et al., 2006*; *Stanford et al., 2010*; *Stanford and Salinas, 2021*; *Gladwell, 2005*; *Gigerenzer, 2007*). In these decisions, accuracy does not increase when more time is available. For example, rats can reliably categorize odors based on just one sniff (about 300 ms), and more sampling time leads to no further improvement. This is not due to the simplicity of the task and holds equally for both simple and complex

odor-discrimination tasks (*Uchida et al., 2006*; *Uchida and Mainen, 2003*; *Zariwala et al., 2013*). Archerfish, the subject, of the present study, monitor the speed, direction, and height of falling prey and rapidly turn towards the corresponding ballistic impact point (*Schlegel and Schuster, 2008*; *Rossel et al., 2002*). In this decision, no relation was found between the average response latency and the accuracy of the turns (*Schlegel and Schuster, 2008*). Humans can reliably classify natural visual images after surprisingly brief glimpses (of only 20–40 ms, see *VanRullen and Thorpe, 2001*; *Bacon-Macé et al., 2005*; *Uka and DeAngelis, 2003*; *Harwerth et al., 2003*; *De Bruyn and Orban, 1988*; *Snowden and Braddick, 1991*; *Nachmias, 1967*; *Tulunay-Keesey and Jones, 1976*) and more recent approaches demonstrated that the perceptual parts of human decision making can often be much faster than previously thought (*Stanford and Salinas, 2021*; *Salinas et al., 2019*). Such rapid decisions are certainly common in sports. For example, hitters in baseball have only 400 ms to move their bat to the optimal strike point, a movement that can only be based on about 70 ms of visual processing (*Higuchi et al., 2016*; *Gray and Cañal-Bruland, 2018*; *Kidokoro et al., 2019*). Finally, even slow, and probably complex human decisions, such as which product to buy in a store, can sometimes deviate from the expected speed-accuracy tradeoff and be superior when made with lesser time and lesser information (*Gladwell, 2005*; *Gigerenzer, 2007*; *Dudey and Todd, 2001*; *Beilock et al., 2004*; *Gigerenzer and Brighton, 2009*).

Despite these deviations, there is of course no doubt that most decisions do show speed-accuracy trade-offs. But why is this type of decision much more common and why are not simply all decisions both fast and accurate? One reason could be that the decisions that lack a speed-accuracy trade-off manage to be so fast because they profit from using simplifying heuristics (*Gladwell, 2005*; *Gigerenzer, 2007*). As we illustrate with a specific example below this allows being fast and accurate

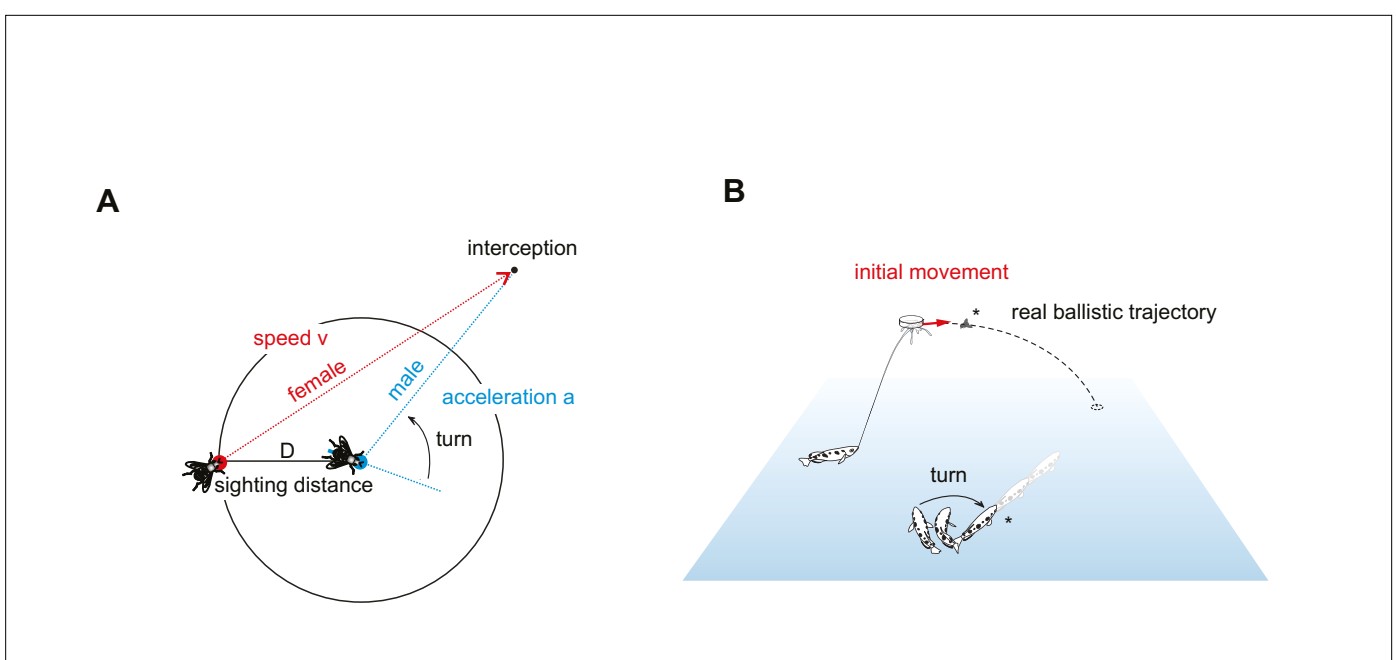

**Figure 1.** Two highspeed decisions are used by animals to catch objects. (**A**) Male hoverflies use a rapid open-loop strategy to catch passing females: When sighting a female that moves at speed 'v,' at sighting distance 'D' from the male, then the male rapidly turns so that a constant acceleration 'a' will automatically make him arrive at the point of interception at the appropriate time. In this decision, the male takes information only once and uses hardwired circuitry that assumes fixed values for *D*, *v*, and *a*. While this simplifies the complex interception considerably it would make it impossible to intercept targets with other values of *D*, *v*, or *a* (*Collett and Land, 1978*). (**B**) Archerfish down aerial prey with shots of water and also use an open-loop strategy to arrive simultaneously with their prey as soon as it hits the water surface. The responding fish, shooters, and bystanders alike, sample visual information on the speed, height, and direction of the initial falling motion and turn right towards where ballistically falling prey would later land. These turns are part of a maneuver called C-start (*Schuster, 2023*; *Wöhl and Schuster, 2007*), that also lends the fish the level of speed that, when kept, make it arrive just in time. Depending on visual contrast, water temperature, height of prey, and the distance from the future landing point, fully accurate turns can be initiated in as little as 40 ms (*Schlegel and Schuster, 2008*). In contrast to the conditions in the male hoverfly, the ecological constraints in archerfish do not allow similar simplifying assumptions and archerfish can respond appropriately from a large range of distances and initial orientation and when encountering targets of unusual size or speed (as will be shown below).

for specific conditions that enormously reduce the complexity of the problem. The decisions would, however, not be useful under more variable conditions as they lack the flexibility to adjust to these other conditions. In other words, the flexibility to cope with changing conditions could be what distinguishes the more common slower decisions (i.e. those with speed-accuracy tradeoffs) and the more unusual highspeed decisions. The evidence that is presently available – from the movement of baseball outfielders and frisbee-catching dogs (*McBeath et al., 1995*; *Shaffer et al., 2004*) to the chase of hoverflies (*Collett and Land, 1978*) – seems to support this view (also see *Gladwell, 2005*; *Gigerenzer, 2007*). The point is perhaps best illustrated by the decision a male hoverfly must make to intercept a passing female (*Figure 1A*). As soon as a female (traveling at speed $v$) comes into sighting distance ($D$), the male rapidly decides on a turn angle that – when the male afterwards accelerates at its maximum rate ($a$) – will lead him right to the later point of interception. An elegant series of experiments showed that this very rapid and accurate decision is based on hardwired assumptions about the speed $v$ and size (and hence detection distance $D$) of a typical female hoverfly and about the typical maximum rate $a$ of the male's acceleration. These variables appear all to be genetically imprinted in the neural circuits that govern the male turn decision. Expectedly, challenging the males with objects that deviated from these assumptions, caused them to make robust errors that were predicted based on a simple heuristic model that the fly appears to use (*Collett and Land, 1978*). The turn decisions of male hoverflies are thus an

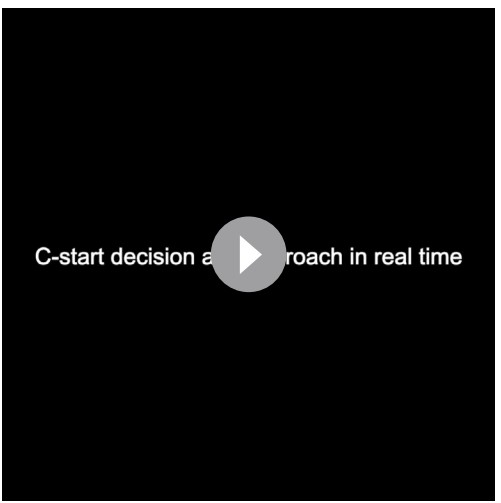

**Video 1.** Example of a turn decision made with a real falling object in real speed and slowed down. Movie from the experiments of *Figure 3C–G* ('Real'), showing a turn decision made in response to the movement of real ballistically falling prey. Prey was released by an air stream from the top of a non-transparent platform that was located above the tank. To illustrate the speed of the decision, the initial part of the movie shows the turn decision and the subsequent approach to the landing point in real time. Subsequently, a slowed-down version (17-fold) of the same scene is shown. Three fish turn to the later impact point of prey. The red line shows the aim of the fish that responded first. It is determined right before the fish takes off, at the end of its so-called C-start maneuver. The scene is filmed from below at 500 fps. Note that all starts are launched faster than an Olympic sprinter can respond to the start gun. They would, therefore, all be considered false starts in the Olympic Games.

https://elifesciences.org/articles/99634/figures#video1

excellent example of a very fast and accurate decision that lacks flexibility but has evolved to work specifically for a restricted set of conditions. This is all that is ever required in the specific behavioral context (*Collett and Land, 1978*).

Here, we examine the degree of flexibility in another highspeed decision that lacks a speed-accuracy tradeoff and in which the ecological context might favor more flexibility. This decision is an essential part of the hunting behavior of archerfish, in which the fish shoot down aerial prey with jets of water (*Dill, 1977*; *Schuster, 2018*; *Gerullis and Schuster, 2014*; *Figure 1B*). Once a prey insect is successfully dislodged, shooters and bystander archerfish have only the little time that remains till prey hits the water surface to turn and dash off to be at the impact point of prey just when it arrives there. In the wild, missing this point or not being there in time, would typically mean losing prey to the more numerous competing other surface-feeding fish (*Rischawy et al., 2015*). The solution archerfish have evolved to secure their prey and to outperform their competitors is to watch the initial movement of dislodged prey and then quickly and accurately select the appropriate turn and take-off to be at the later landing point at the right time (e.g., *Schlegel and Schuster, 2008*; *Rossel et al., 2002*; *Schuster, 2023*; *Figure 1B*). Their so-called predictive C-start decisions are among the fastest C-starts known in fish and are initiated much faster than an Olympic sprinter can respond to the start gun (*Sillar et al., 2016*; see *Video 1* for an appreciation of the decision's speed).

An essential feature of the decisions – and the basis for the present study – is introduced in *Figure 2*. It was discovered in experiments in which archerfish typically faced real ballistically falling

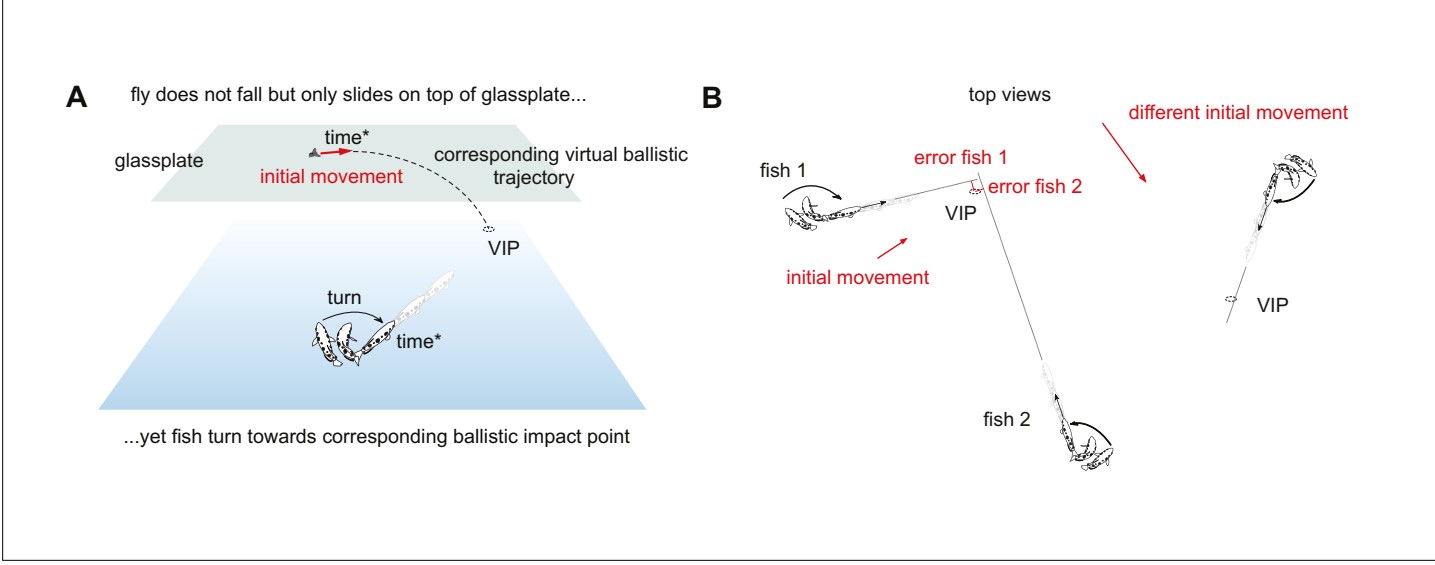

**Figure 2.** When prey is prevented from falling, archerfish turn to the virtual ballistic impact point. (**A**) Sketch of an experiment in which archerfish are occasionally challenged with prey that is prevented from falling but that only slides horizontally on a glass plate. As long as such tests are rare, archerfish still elicit turn decisions even though the fly does not fall and will never ever come down. Remarkably, the aim of these starts is towards the appropriate 'virtual' impact point (VIP). To show this, one uses ballistics, to calculate for each test with a sliding fly where the fly would have landed, given its initial values of motion (initial horizontal speed, direction, and height). Using the results accumulated from many such tests shows that, for any combination of the initial motion values, the fish turn toward the expected ballistic landing point. The decision circuit thus maps the combination of initial motion values to a turn towards the expected ballistic landing point. (**B**) Sketch to show how the accuracy of turns toward the appropriate VIPs can be quantitatively assayed for each of the various combinations of sliding speed, direction, and height. An ecologically relevant measure of the turn's accuracy is how close the chosen orientation would bring the fish to the VIP, i.e. how close following the chosen initial direction would bring the fish to the point of reward. The sketch also illustrates that this measure allows pooling responses with widely different initial conditions, i.e., wide differences in the initial distance and required turn angle for each VIP. (**A, B**) This experiment provides the basis of all experiments in this paper. It shows that turns can (i) be elicited by purely horizontal motion, and that (ii) these turns are towards the virtual ballistic impact points, i.e. the points predicted from initial motion, using ballistics.

The online version of this article includes the following figure supplement(s) for figure 2:

**Figure supplement 1.** The accuracy of the archerfish turn decisions can be precisely quantified.

prey but were occasionally challenged with prey that was prevented from falling (and only slid on top of a glass plate; *Figure 2A*; *Rossel et al., 2002*). When many such interspersed rare cases with sliding flies were later analyzed, it turned out that fish still responded, although nothing fell. Moreover, their response latency – measured between the onset of prey motion and the onset of the turns – was the same as in most tests in which real prey was falling (*Rossel et al., 2002*). Given that there never was an actual later impact point, where did the fish turn to when nothing was actually falling? This became clear when the *virtual* ballistic impact points were calculated for each individual test made with sliding flies. Using simple high school physics (and confirming that the prey fell accordingly, *Rossel et al., 2002*) allowed to calculate the virtual landing point for each combination of initial height, speed, and direction that the sliding fly had in the interspersed tests. This combination differed from one test to the next. The accuracy of each turn decision with respect to each calculated virtual landing point could then be measured in the same way as with real falling objects (*Figure 2B*). This analysis revealed that for each set of conditions the fish had turned to the appropriate virtual impact point, calculated based on ballistics, just as they would toward a real later impact point of real falling prey (*Rossel et al., 2002*). In other words, based on initial prey movement the fish were able to turn toward the appropriate virtual ballistic landing points. This implies that the fish were able to map the initial values of motion to the turn angle they needed to produce and that this mapping is appropriate for ballistically falling objects (*Figure 2B*).

In the glass plate experiments of *Figure 2* prey movement was not controlled by the fish but flies were made to slide by blowing an airstream at them. This introduces another convenient aspect of the decision that we will exploit throughout the study: The archerfish's turn decisions exclusively use

information that is sampled during a brief interval after onset of prey motion (*Schlegel and Schuster, 2008*; *Reinel and Schuster, 2014*; *Reinel and Schuster, 2018a*; *Reinel and Schuster, 2018b*). This information even includes the initial height from which prey is falling (*Reinel and Schuster, 2018a*). In other words, all experiments available have shown that the start decision does not use or has no access to prior information. Instead, it relies exclusively on information sampled once prey starts moving.

With this background, it may now become clear what makes the turn decisions of archerfish so interesting for studying the limits in flexibility in highspeed decision making. Because the turn decisions are appropriately made to the virtual ballistic landing point of prey that is not even falling, it is clear that the fish's turn is somehow tuned to ballistically falling prey. This allows the fish to turn to the appropriate virtual landing point for each given combination of initial values of prey motion (*Figure 2*). Because all archerfish studied so far in the lab and in the wild consistently showed robust turn decisions towards the inferred ballistic landing points (*Schlegel and Schuster, 2008*; *Rossel et al., 2002*; *Rischawy et al., 2015*; *Reinel and Schuster, 2014*; *Reinel and Schuster, 2018a*; *Reinel and Schuster, 2018b*; *Wöhl and Schuster, 2006*; *Wöhl and Schuster, 2007*), we expected the decision to be hardwired to ballistics. Hardwiring the decision circuit to ballistic falling patterns could be fixed genetically and/or established during the early ontogeny of the fish. A striking advantage of yet more flexibility is also not immediately obvious: While an initial aim to the ballistic landing point would be in error for non-ballistically falling prey, this initial error of the fish's turn could always be corrected during the actual approach trajectory. Nevertheless, we demonstrate here that the decision is not hardwired to ballistic falling. We show that adult archerfish are capable of learning to extract and to use a novel non-ballistic rule of how to connect the three independent input variables of prey motion to the point to which they must turn to receive a reward.

## Results
### Testing a virtual reality approach to study the flexibility of the decision

Experiments with purely virtual impact points and their precise quantitative analysis (as introduced in *Figure 2A, B*) would, in principle, offer a way of testing whether the archerfish's highspeed decisions are limited to ballistically falling prey. For any given combination of initial speed, initial height, and direction of prey movement and for any given orientation and position of the responding fish, ballistics requires the fish to turn by a specific angle. Now suppose the relation between initial prey movement and the point to which the fish must turn would be changed to no longer be defined by ballistics, but by some other rule. Then all turn decisions after the switch would be systematically in error and the key question is whether at least some fish would eventually be able to adjust their start decisions accordingly. Exploring this requires a virtual reality (VR) system. Because the fish need to be rewarded at the appropriate place and time to stay motivated over extended series of experiments (that last many months), it is not at all clear, if such a system could work without introducing additional cues. To be useful for the question at hand, the system needs to fulfill four conditions: (i) First, it must present the information with sufficient spatial and temporal resolution as needed by the decision circuit. (ii) Second, it needs to present the reward at the positions and times as required by ballistic or arbitrary other rules but without adding further information that could interfere with the decision. (iii) Third, the VR system would have to work so well that there is no statistically detectable difference between virtual and real responses. (iv) Finally, the system must keep the fish motivated for the many months of experimentation during which the fish will exclusively be facing virtual objects. Clearly, if only one of these conditions is not met, then the chosen VR system would not be useful to explore our fundamental question of whether the decisions are or are not flexibly adjustable to non-ballistic relations.

The first step in our endeavor, therefore, was to explore and quantitatively test the suitability of potential setups. After several unsuccessful attempts – including LED-based arrangements used in earlier work (*Strauss et al., 1997*; *Schuster et al., 2002*; *Reiser and Dickinson, 2008*) – we found a system that completely fulfilled all four crucial conditions. In the following, we present evidence for that successful system (*Figure 3*). The setup worked, surprisingly, with an LCD screen at a frame rate of 120 frames per second and employed custom-made automatic feeders that were non-transparent and provided food at the right time and place, coupled to the motion shown on the screen. To critically test the approach, the automatized feeders presented food at the ballistic virtual impact points and

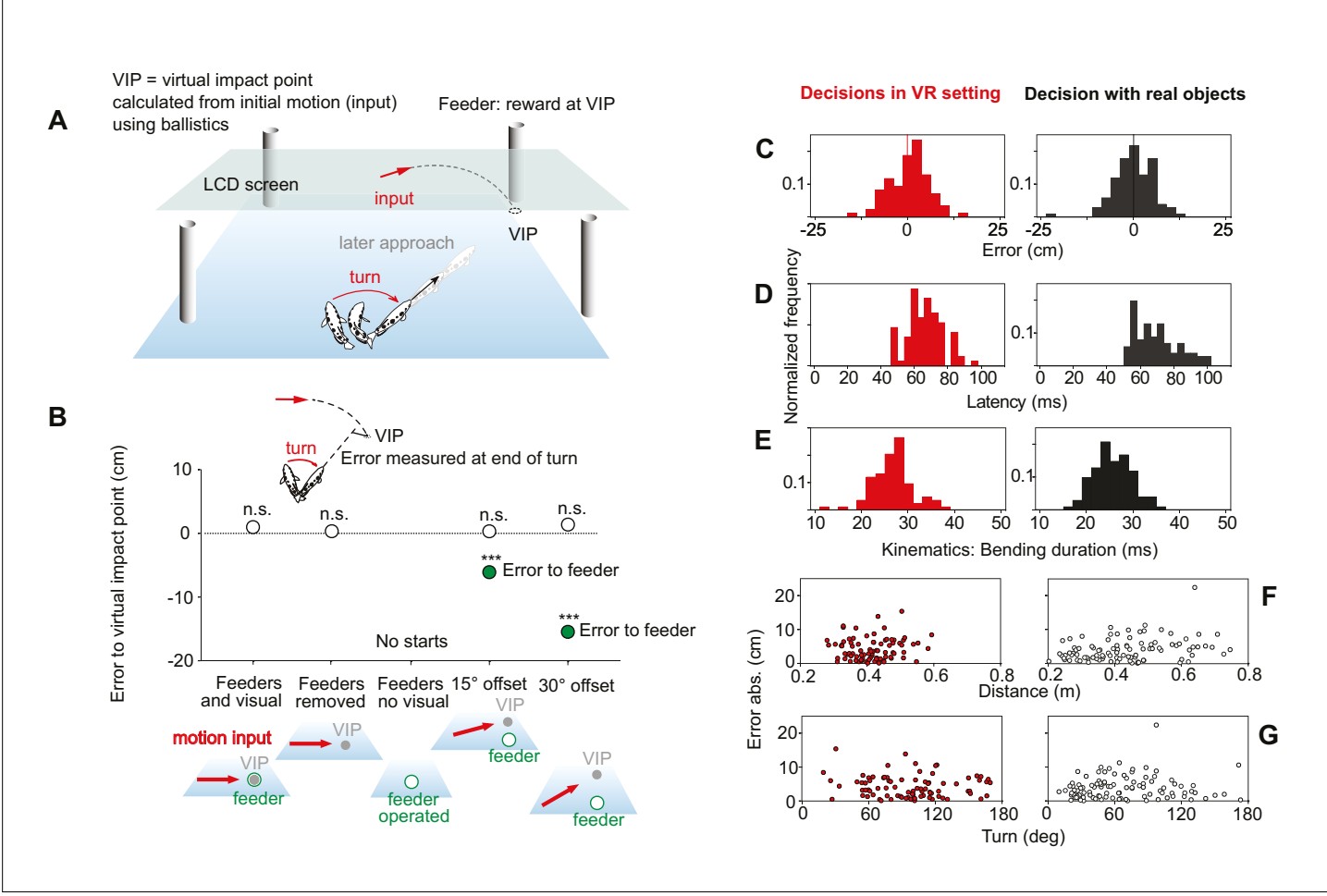

**Figure 3.** Critical tests that directly show the suitability of the virtual reality setup that is used in the present paper. To exclusively work with virtual stimuli, as required in the present study, a system was needed in which the fish would respond to virtual stimuli as in the natural situation. (**A**) Sketch of the successful setup with motion presented on an LCD screen and feeders operated to deliver food in time at the virtual impact points (VIPs), the points at which ballistically falling prey would impact, given its initial motion (see *Figure 2*). (**B**) Evidence that the decisions are not influenced by the presence and operation of the conspicuous feeders: Same errors to VIPs with feeders present or removed. No rapid turns are elicited when feeders are operated without movement shown on the screen. In tests in which the feeders are not at the position of the VIPs but offset from them (by either 11.2 or 22.3 cm), turns minimized the error to the VIPs but not to the feeder positions. Data are represented as medians, inset recalls how errors were defined (see *Figure 2* and *Figure 2—figure supplement 1*). (**C–G**) Direct comparison of the turn decisions with real and virtual impact points. Same errors (**C**) to virtual and real impact point, same response latency (**D**), same kinematics (**E**, duration of bending phase). (**F, G**) As with real objects, accurate starts to virtual landing points are also possible across wide ranges of distance (**F**) and turning angle (**G**). See *Figure 2—figure supplement 1*, *Figure 3—figure supplement 1* for additional information, *Videos 1 and 2* for examples showing the turn decision, and how they were evaluated. See text for detailed statistics. \*\*\*p<0.001, n.s. not significant.

The online version of this article includes the following source data and figure supplement(s) for figure 3:

**Source data 1.** Contains all data of *Figure 3*.

**Figure supplement 1.** The apparatus that passed all critical tests (*Figure 3*) for studying the archerfish turn decisions under virtual reality (VR) conditions.

at the appropriate time. First, the setup readily and continuously elicited turns toward the calculated ballistic impact point (*Figure 3B–G*). Next, a series of experiments showed, that the highly conspicuous feeders did not influence the decision at all but that the decision was still exclusively based on the initial target motion shown on the screen (*Figure 3B*): Operating the feeders without showing motion on the screen failed to elicit turn decisions. Most importantly, when feeders were placed not at the ballistic virtual impact points (VIPs) but systematically offset from them, then turns were aimed at the VIPs and not at the feeders – even though this was where the fish got their reward (*Figure 3B*;

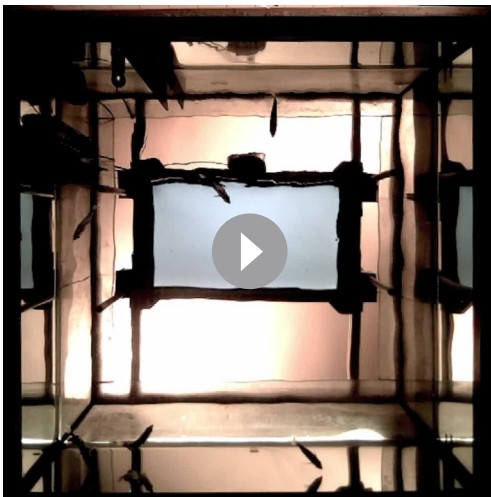

**Video 2.** Example of a turn decision with a purely virtual impact point. Movie from the experiments of *Figure 3*, now with only a virtual impact point. The movie shows a turn decision made in response to the movement shown on the LCD screen. The scene is filmed from below at 500 fps and is shown slowed down 17-fold. It shows one of the interspersed tests in which prey motion was not towards a feeder so that the inferred virtual impact point was offset from the positions of the feeders. The region around the virtual impact point (i.e. where real ballistically falling prey with the same initial speed, direction, and height would land) and the region around the feeder are indicated. Note that the fish turns to the virtual impact point and not to the feeder. The red line shows the fish's aim at the end of the final straightening phase (so-called stage 2) of the C-start maneuver.

https://elifesciences.org/articles/99634/figures#video2

15° offset, n=353; 30° offset, n=198). These tests were conducted so that the VIPs also differed relative to other potential landmarks in the tank and so they showed that the fish were also not starting to positions marked by landmarks other than the feeders. In other words, even in the presence of the feeders were the decisions only based on the movement information that was presented on the screen and were not influenced by cues from the feeders or from any other landmarks in the tank.

Next, we needed to critically examine if the decisions made to ballistic VIPs (*Video 2*), as mimicked and rewarded by the system, were equivalent to those made with real falling objects (see *Video 1*). Given the limitations in the temporal resolution of the screen it is striking that no aspect of the decisions was statistically different under real and virtual conditions (*Figure 3C–G*, n=101 (real) and n=85 (virtual) starts). The accuracy of the aim achieved at the end of the turns (measured as introduced in *Figure 2*) was not statistically different (p=0.506, Mann-Whitney; *Figure 3C*) and always minimized the error to the impact point, regardless of whether it was real or virtual (difference from zero: p=0.903 (real), p=0.455 (virtual); One-Sample Signed Rank) with also no significant difference in variance (p=0.589, Brown-Forsythe). The turns also had the same latency (between onset of prey motion and onset of the turns; p=0.297; Mann-Whitney; *Figure 3D*) and kinematics (*Figure 3E*: no difference in bending duration under real and VR conditions, p=0.071, Mann-Whitney). Furthermore, responses came from a large range of orientations and distances and, most importantly, were equally accurate across all distances (*Figure 3F*) and turn sizes

(*Figure 3G*). In summary, despite the limited frame rate of the screens and despite the need to use highly conspicuous feeders in the majority of the (rewarded) presentations, the turn decisions were equivalent in all examined aspects (accuracy, latency, kinematics) to those obtained with real falling objects. In other words, the setup was sufficient to start our endeavor and allowed to study the archerfish's highspeed decision exclusively with virtual stimuli.

## Challenging the decision with a new rule that links input and rewarded output

The VR setup fulfilled all conditions required: it robustly elicits turn decisions that deviate in no detectable way from those elicited with real falling objects and that enable the fish to turn to the later ballistic landing point, real or virtual, from a wide range of distances and for all required turn sizes. This allowed us, to use the VR setup to explore the central question of the present study: would the decision be capable of adjusting to a novel rule of how the input variables allow to infer the rewarded virtual impact points? If the fish continued using ballistics to infer the VIP, as predicted, then their starts would now always be in error. The fish could then still learn to efficiently correct their initial errors later during their actual approach trajectory. *Figure 4A* introduces the new rule that we chose to face the fish with. It generates a systematic and substantial deviation (of 14.9 cm) between the new VIPs and the ballistic VIPs as inferred from the initial movement. After having traveled straight throughout what was the previous decision time, the movement direction is deflected by 39.8° (to the left, s. Methods),

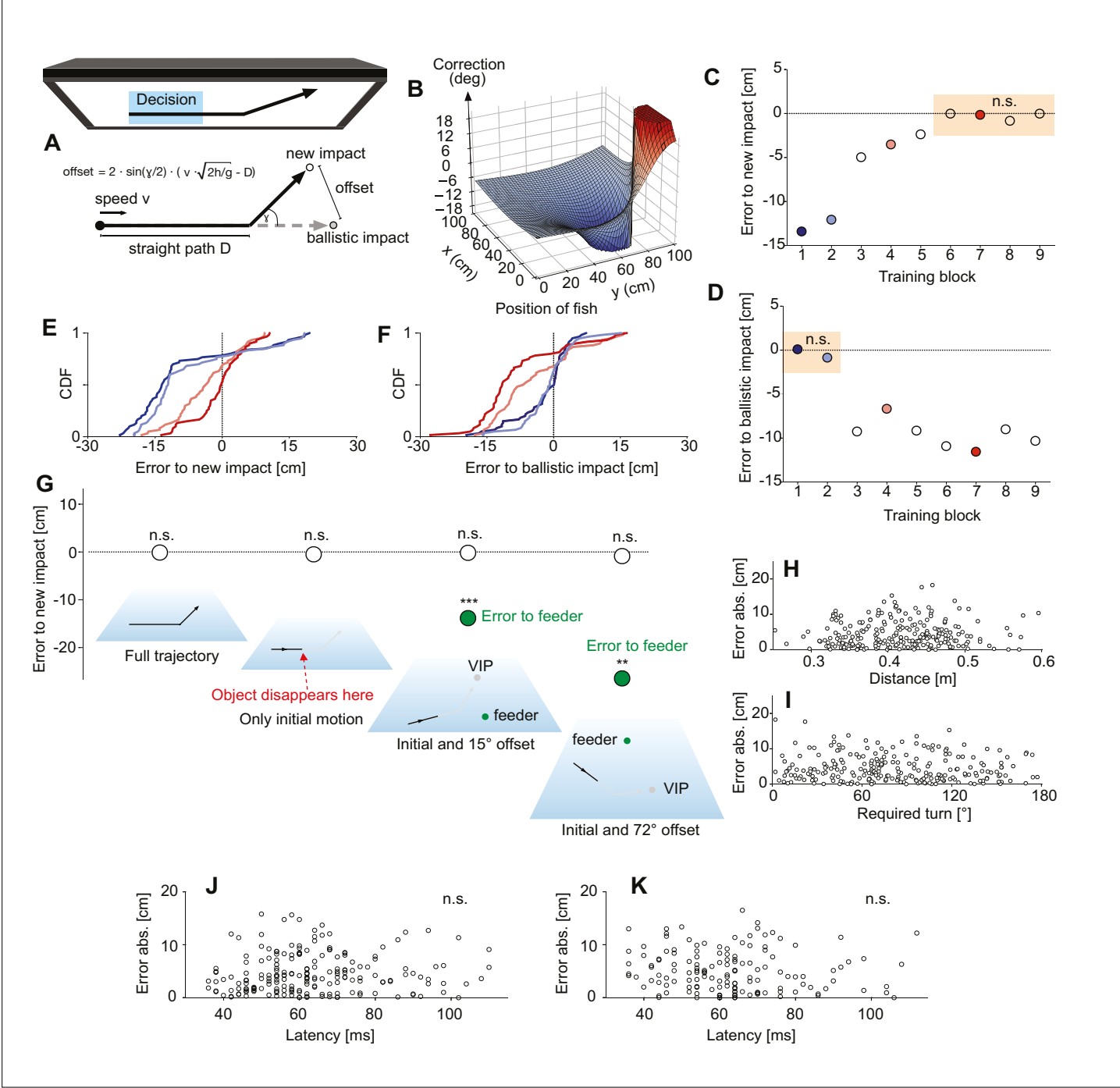

**Figure 4.** The highspeed decision is not hardwired to ballistics but can be trained to a new rule. (**A**) Sketch of the movement pattern used to introduce new virtual impact points (VIPs) that are still strictly defined by initial motion but systematically differ from the ballistic VIPs. Trajectories were initially straight but – after the time it typically takes the fish to make their decision – then changed their direction as illustrated. Once this new movement pattern was introduced, reward was given exclusively at the corresponding new 'deflected' VIPs and no longer at the ballistic VIP. (**B**) Learning to adjust the turns to initial prey motion requires non-trivial corrections to the previous turning angles (to the ballistic VIPs). These corrections depend on the fish's initial position and a different map is needed for each set of motion input variables (as illustrated in *Figure 4—figure supplement 1*). (**C, D**) Median errors of the turns relative to the new (**C**) and to the old ballistic (**D**) VIPs in successive training blocks 1–9 (each with 60 evaluated turn decisions), with food always delivered at the new VIPs. Colored background highlights when turns were oriented towards the new (**C**) and when to the old ballistic (**D**) VIPs. (**E, F**) Cumulative density functions (CDFs) at the blocks indicated by colored circles in (**C, D**) to show how the systematic error to the new VIPs systematically decreased (**E**), whereas errors to the ballistic landing points increased (**F**). (**G**) Errors made in several critical tests – from left to right: the deflection was visible and feeders were present at the new VIPs, just as during training (n=240), interspersed tests with only the initial straight trajectory

*Figure 4 continued on next page*

*Figure 4 continued*

shown (i.e. prey vanished before the deflection would occur) (n=68), interspersed tests (rate: 1 in 8) with also only the initial straight trajectory shown but direction of motion offset from the direction of the feeders by either 15° (n=197) or by 72° (n=145) to test if fish turned to the new VIP or to the feeder positions. (**H, I**) After training the fish were able to again respond appropriately over a wide range of distances (**H**) and required angles of turn (**I**). (**J, K**) Absence of speed-accuracy trade-off of the re-programmed highspeed decisions. Plot of magnitude of error in aim versus response latency in the critical tests shown in (**G**) with only initial pre-deflection prey trajectory and offset of either 15° (**J**) or 72° (**K**). Note the absence of significant correlation between error magnitude and response latency (p>0.3; R² <0.005). Data are represented as median in (**C, D, G**). ***p<0.001, **p<0.01, n.s. not significant.

The online version of this article includes the following source data and figure supplement(s) for figure 4:

**Source data 1.** Contains all data of *Figure 4*.

**Figure supplement 1.** Learning the new rule (*Figure 4*) requires complex position-dependent corrections, that differ with every different input constellation.

and the reward is now presented at the appropriate 'deflected' VIP. According to this new rule, the connection between initial movement and the point of reward is still precisely defined. It would, therefore, in principle, be possible to extract and use this new rule to infer the point of reward from the initial pre-deflection movement. Specifically, it would be easy to critically test whether the fish (or at least some of them) were using the new rule, by testing them exclusively with the initial movement (i.e., without the later deflection), and to measure if the turns are directed towards the calculated deflected or towards the calculated ballistic VIPs. Because the model behind our new rule might appear simple it is important to stress that learning to turn according to this new rule is not at all simple. It would require the fish to learn non-trivial corrections that can be appreciated by considering *Figure 4B*: Relative to the turn they had previously made for a given input constellation they would need to add a correction that is not constant but depends sensitively on the position of the responding fish. This map of corrections is, furthermore, not the same for all input constellations. Instead, a new map of the kind shown in *Figure 4B* is required for every combination of input variables (as illustrated with a simple example in *Figure 4—figure supplement 1*).

If adult archerfish used a hardwired decision circuit, they could learn to rapidly modify their subsequent approach path, or they could abandon their rapid decisions altogether and instead use slower responses. To examine this, we now exclusively challenged the fish for many weeks with the new deflected trajectories and rewarded them at the newly defined VIPs. During this time, we continuously sampled the occurrence and error of all turn decisions (*Figure 4C, D*). Interestingly, the fish never ceased to respond with their characteristic turns (response probability remained constant across all training blocks, p>0.08, Pairwise Chi-square). The kinematics of the turns and latency also remained constant (bending and straightening phase of their C-starts both p>0.28, latency p>0.07; Kruskal-Wallis). Furthermore, their aims were systematically in error for the initial training blocks (deviation from zero error to the deflected VIPs p≤0.001, One-Sample Signed Rank) as expected (*Figure 4D*; deviation of aim to ballistic VIPs from zero error p>0.06; One-Sample Signed Rank; *Video 3*). However, gradually the errors made to the new 'deflected' VIPs became

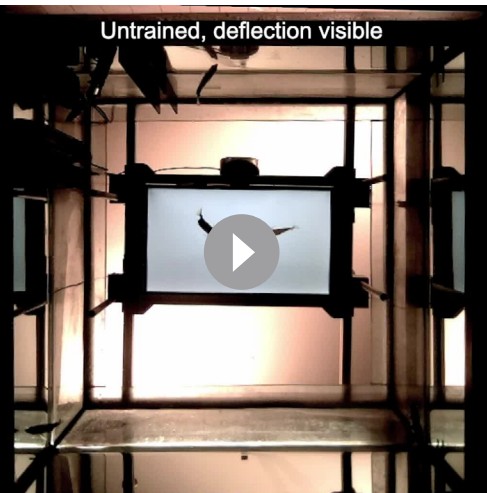

**Video 3.** Example from the start of training when fish still turned based on ballistics. At the initial stage of training to the new rule (*Figure 4*), turns were towards the ballistic virtual impact point. The scene is filmed from below at 500 fps and shows slowed down 17-fold. The movement of the disk-shaped prey object and its deflection after an initially straight path is visible on the LCD screen above the tank. A feeder automatically delivers food at the deflected landing point. The aim of the turn (at the end of C-start) of the responding fish is indicated as a red line. Also indicated are the region around the feeder and around the inferred virtual ballistic impact point.

https://elifesciences.org/articles/99634/figures#video3

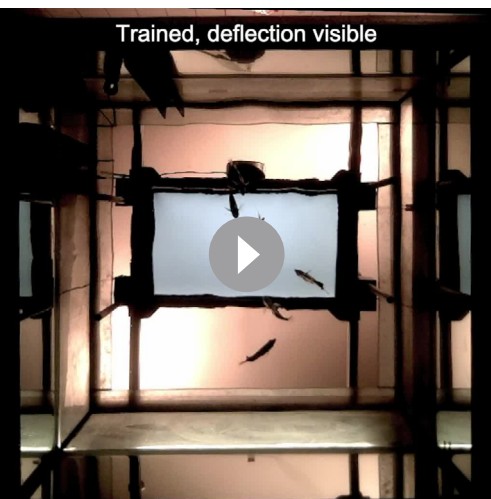

**Video 4.** After training, turns are no longer towards the inferred ballistic impact point. Movie from late stage of training to the new 'deflected' falling rule (*Figure 4*; block 7 of training). The scene is filmed from below at 500 fps and is shown slowed down 17-fold. Same movement of disk with visible deflection shown on LCD screen above the tank as in *Video 3*. The turns of all responding fish are now toward the new deflected virtual landing point, where food is also delivered during training. The red line indicates the aim at the end of the C-start turn of the first responding fish. Although only the turn decision of the first responding fish enters the analysis, a yellow line illustrates the aim of the fish that was second to respond. At this stage of training, all fish no longer turned to the ballistic virtual impact points but to the 'deflected' impact points.

https://elifesciences.org/articles/99634/figures#video4

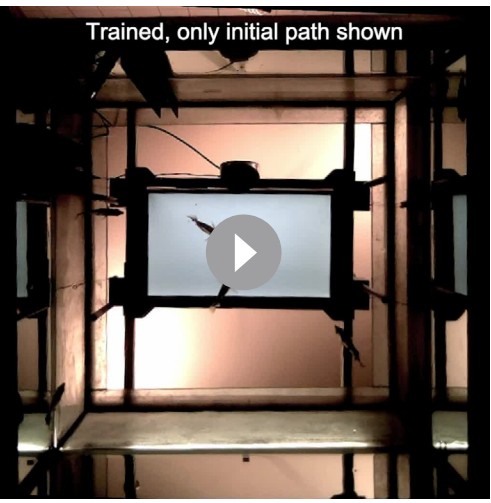

**Video 5.** Example of a critical test to show that the decision had acquired a new rule for inferring the impact point. Example of an interspersed test to examine what the fish had learned during the training in *Figure 4*. Only the initial straight path of the prey's trajectory is shown, i.e. prey vanishes before the deflection. Furthermore, the direction of the initial path was so that the deflected virtual impact point would be offset from the position of a feeder. The regions around the expected ballistic landing point (Ballistic), the expected deflected landing point (Deflected) and food release (Feeder) are highlighted. The aim at the end of the fish's C-start is highlighted by the red line. The scene is filmed from below at 500 fps and is shown slowed down 17-fold. Prior shooting behavior illustrates that shooting behavior can be used to confirm that the fish were always motivated to hunt throughout the experiments.

https://elifesciences.org/articles/99634/figures#video5

smaller and, after more than 1750 presentations (55 experimental days), the turns clearly minimized the error to the appropriate new 'deflected' VIPs (*Figure 4C, E, F*; *Video 4*) and no longer to the ballistic ones.

This surprising finding does by no means imply that the decision has now been based on the new rule. For example, the fish could have learned to feed information about the deflection of the prey's trajectory into their unfolding turns and so achieve their new final turning angles, even though they had initiated their turns before the deflections were visible. To test this possibility, we interspersed tests in which the trained fish were only shown the initial linear part (i.e. the moving prey disappeared prior to the deflection) among many presentations with the full trajectory (including the deflection). The errors made in the interspersed tests (n=68) were not different from those made when the full trajectories were visible (*Figure 4G*). This shows that in these critical tests, the starts were still only made using the initial motion and did not require visibility of the deflection. Another way to explain the finding, however, could be that now the fish have learned to take the feeder positions into account, even though they did not in the previous experiments (*Figure 3B*). In this scenario, the fish could have learned to use initial movement as an indicator which feeder to approach. To examine this possibility, we interspersed tests in which the direction of initial target movement was always either 15° or 72° off from the feeder position and in which, additionally, exclusively the initial movement was shown (i.e. prey vanished before the deflection). Furthermore, feeders were always offset both from the ballistic as well as from the new VIPs. Yet, the turns the fish made in these tests were still aimed at the new deflected VIPs (differences from zero error p=0.721 at 15° (n=197) and p=0.113 at 72° (n=145); One-Sample t-tests) but not at the feeders and also not at the ballistic VIPs (Kruskal-Wallis: p≤0.001 at 15°;

p=0.005 at 72°; Dunn's p<0.05 for comparison of errors; *Video 5*). Furthermore, the turns – made when only the linear initial motion was visible and when the VIPs were spatially offset from the feeders – did not differ in latency (p=0.686, Kruskal-Wallis) nor in kinematics (bending duration: p=0.242, Kruskal-Wallis) from the turns made without offset and with the full trajectory visible (n=240).

Finally, the new turns were not restricted to just a few positions and orientations of the responding fish. Rather, they were possible and equally accurate across a range of distances and turn angles (*Figure 4H, I*). We also demonstrate another important feature, the absence of a speed-accuracy trade-off also in the re-programmed decisions. The absence has previously been described for real, ballistically falling prey (*Schlegel and Schuster, 2008*). In *Figure 4J, K* we plot the magnitude of the error in the aim (immediately at the end of the C-starts, see *Figure 2B*, *Figure 2—figure supplement 1*) against response latency for the critical tests of *Figure 4G*, in which only the linear initial part of motion was shown and offset by either 15° or 72° from the direction of the feeders. For both plots, there was no significant linear (p=0.318, $R^2$=0.005, *Figure 4J*; p=0.469, $R^2$=0.004, *Figure 4K*) or Spearman correlation (p=0.128 and 0.174, respectively, in *Figure 4J, K*). Our findings, therefore, demonstrate that a highspeed decision that lacks a speed-accuracy tradeoff can be flexibly reprogrammed to a new rule.

## Immediate generalization to untrained input constellations

We next wondered about the degree of abstraction, if any, that was involved in reprograming the decision. This can be examined conveniently because all training to the new rule was carried out exclusively with targets presented at only one height level. In addition, possible target speed levels were limited, and prey was of only one size. In principle, the decision network could be retrained by simply replacing the angle required in the previous turns, given the position of the fish and the initial motion of prey (speed, direction, height) with the new angle. Formally this could be described as simply overwriting a 'lookup table.' In this scenario the fish would not detect the underlying rule in a way that would enable it to generalize appropriately to novel untrained settings. So, in initial tests with motion shown at untrained height levels, predictably wrong turns should occur, because a lookup table would then either only contain (i) the former entries that were appropriate for ballistic motion (which we will call the prediction 'ballistic'), or (ii) all its former ballistic entries could be substituted with the entries that were appropriate at the training height (which we will call the prediction 'substitute'). Both predictions can be tested quantitatively, by calculating the turns the fish would have to make when they first encounter targets at untrained height levels (see *Figure 5A* for an illustration of the predictions). For the actual tests, the untrained height levels were chosen such that the predicted errors to any of the correct VIPs would be at least 6 cm (and thus readily detectable). The actual errors made relative to the appropriate actual VIP are shown for the first 30 turns made with the novel larger (*Figure 5B*) or novel lower (*Figure 5C*) height level. Interestingly, the turns already minimized the error to the VIPs that were appropriate for the untrained novel height levels. They were appropriate right from start with no trend that errors would decrease (linear regression: all p>0.1). A detailed analysis of the errors (*Figure 5D, E*) showed that the turns minimized the error to the deflected landing point also at the untrained height levels (difference from zero: both p>0.07; One-Sample t-test). They were also not directed to the ballistic landing point (prediction 'ballistic;' both p<0.001; One-Sample Signed Rank) and not towards the deflected landing point at the training height (prediction 'substitute;' both p<0.001; One-Sample Signed Rank). Again, none of the characteristics of the turns had changed in the novel situation compared to those after training (also n=30 right before these tests): latency: p=0.098, kinematics: p=0.177 (Kruskal-Wallis), variance of errors: p=0.188 (Brown-Forsythe). In other words, the new rule is not represented by updating a simple lookup table but in a way that allows immediate generalization to the new untrained height levels.

We next tested whether re-programming of the decisions also allows generalization to untrained prey sizes and speed levels. Pursuing the same strategy as illustrated for height (*Figure 5A*), we faced the trained fish with either a new absolute prey size or with new levels of prey speed. Again, the decision circuit could then use the previous entries in the hypothetical lookup table for these sizes and speed levels, that it would have from previous encounters with real, ballistically falling objects. Alternatively, it could have substituted the ballistic values with those that were appropriate during the training in *Figure 4*. Again, the trained fish immediately turned to the deflected landing points (differences from zero error: each p>0.4, One-Sample Signed Rank; difference in aim before and after size

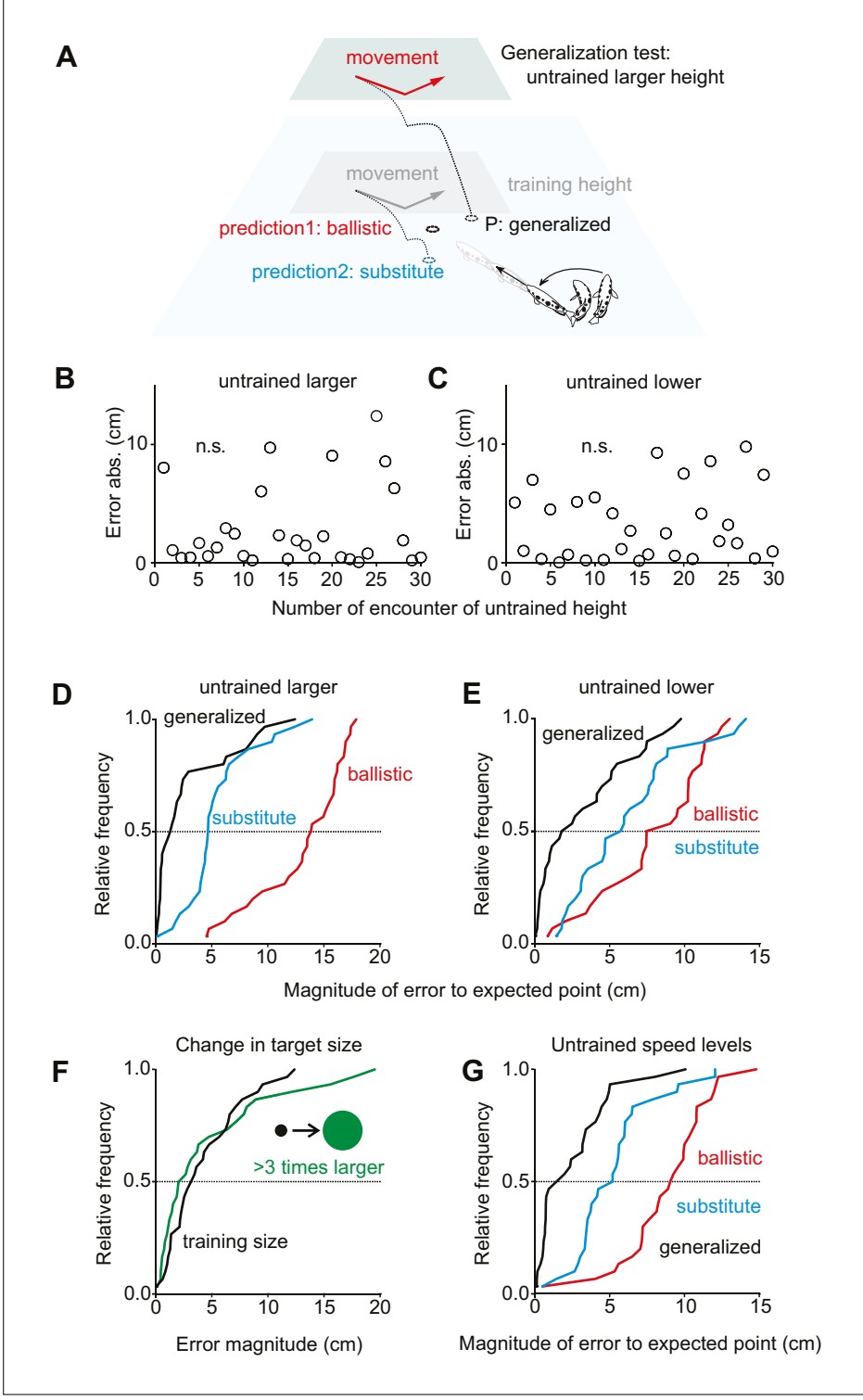

**Figure 5.** The new rule is represented in a way that allows immediate generalization. (**A**) Idea for generalization tests are possible because training to the 'deflected' trajectories (*Figure 4*) was only for one level of target height. When faced with a larger than training height, fish would only be able to turn to the corresponding impact point P if they had acquired a more general rule of how to connect input (height, direction, speed) to the rewarding turn. Had they only exchanged those input-output experiences that were no longer rewarded, then they should still use the previously rewarded input-output relations and turn to the ballistic impact point (red, ''ballistic;'; see text). Had they substituted the old input-output relation for all height levels with the trained new ones, then they

*Figure 5 continued on next page*

*Figure 5 continued*

should turn to point 'substitute' (blue, see text) that would be appropriate for the training height. (**B, C**) When the fish were shown the deflected trajectories at larger (**B**) or lower (**C**) height than experienced during training, then already their first turns minimized the errors to the VIPs that are appropriate for the new rule at the untrained new height level with no indication that errors would initially be large and then decrease. (**D, E**) Closer analysis, using cumulative density functions (CDFs), for the errors made to the predicted points based on hypotheses introduced in (**A**), both for the larger (**D**) and for the lower (**E**) height (both n=30). (**F**) Evidence that learning the new rule had not used prior assumptions on target size. Errors of the turns made in the first 30 tests in which absolute target size was more than three-fold (13 mm) than the target size (4 mm) encountered throughout the training to the new rule. (**G**) CDFs of the first 30 tests with a low target speed (1.425 m/s) that the fish never encountered during training to the new rule. Again, the fish immediately chose turns to minimize the error to the predicted VIP based on the new rule but not the error to the points calculated based on predictions 'ballistic' or 'substitute'.

The online version of this article includes the following source data for figure 5:

**Source data 1.** Contains all data of *Figure 5*.

change p=0.764, Mann-Whitney) and the error distributions were not significantly different (difference in distributions in *Figure 5F*: p=0.278, Brown-Forsythe). Hence, learning did also not involve simplifying a priori assumptions about absolute prey size. The same held for untrained speed levels. When tests with lower than training speed were interspersed among presentations with the speed levels as during training, then already the first 30 turns made at the lower speed (*Figure 5G*) minimized the error to the actual deflected VIP (difference from zero p=0.102; One-Sample t-test), but not to the VIPs given by either ballistics (prediction 'ballistic;' p<0.001; One-Sample Signed Rank) or by using predictions for the deflected VIPs that are based on the trained speed levels (p<0.001; One-Sample Signed Rank). Comparing latency, bending duration, and the variance of errors (speed: last 30 presentations with only training speed; size: last 30 presentations with target size during training) showed that changing target size and speed also did not affect the nature of the turns. Latency and bending duration were not significantly different in the changed versus training conditions (at novel untrained speed: Latency: p=0.280, Mann-Whitney; Stage 1: p=0.076, Mann-Whitney. At novel untrained size: Latency: p=0.305, t-test; Stage 1: p=0.997, Mann-Whitney). Variance was even slightly reduced (and not increased) at the untrained target speed (p=0.021, Brown-Forsythe) and remained unchanged after the increase in target size (p=0.742, Brown-Forsythe). In summary, all tests show that the decisions had been reprogrammed in much more clever ways than predicted. Rather than independently changing the size of an unrewarding turn for each individual constellation – as would seem required because the input variables themselves are varied independently from each other – learning allowed immediate generalization.

## The decision can conditionally operate on two different rules

The fish were now no longer using ballistics to infer the rewarded virtual impact point from initial prey motion. But would they immediately switch back to using ballistics as soon as they encountered objects whose reward point was defined by ballistics, as throughout the fish's previous life? To explore this possibility, we randomly showed our trained fish two differently shaped objects: Either (i) the disk to which the fish responded with turns toward the rewarded deflected VIPs, or (ii) a 'ballistic' object (triangle) that moved straight and for which a reward was given at the ballistic VIP (*Figure 6A*). In the experiments, one of the two objects was chosen randomly to appear in stationary position for 2 s, and then it started moving. Our prediction was that the occurrence of objects with ballistic VIPs would reset everything and that the fish would now only turn towards ballistic VIPs – both for triangles and for circles. However, in experiments to test this prediction, the turns made in response to the 'ballistic object' (rewarded at the ballistic VIP) initially still minimized the error to the deflected VIP (difference from zero error p=0.256; One-Sample Signed Rank; same variance as for disk p=0.343; Brown-Forsythe; first 160 responses to ballistic object) and not to the ballistic VIP point (difference to ballistic landing point: p=0.038, Mann-Whitney; *Figure 6B, C*). However, after 500 additional starts in response to the ballistic objects, turns were equally in error to the deflected and the ballistic VIPs (p=0.648, Mann-Whitney). In contrast, the turns made to the moving disks remained unaffected throughout the same period and remained perfectly matched to the deflected landing point (difference from zero error p=0.273; One-Sample Signed Rank; *Figure 6C*). After about 300 additional

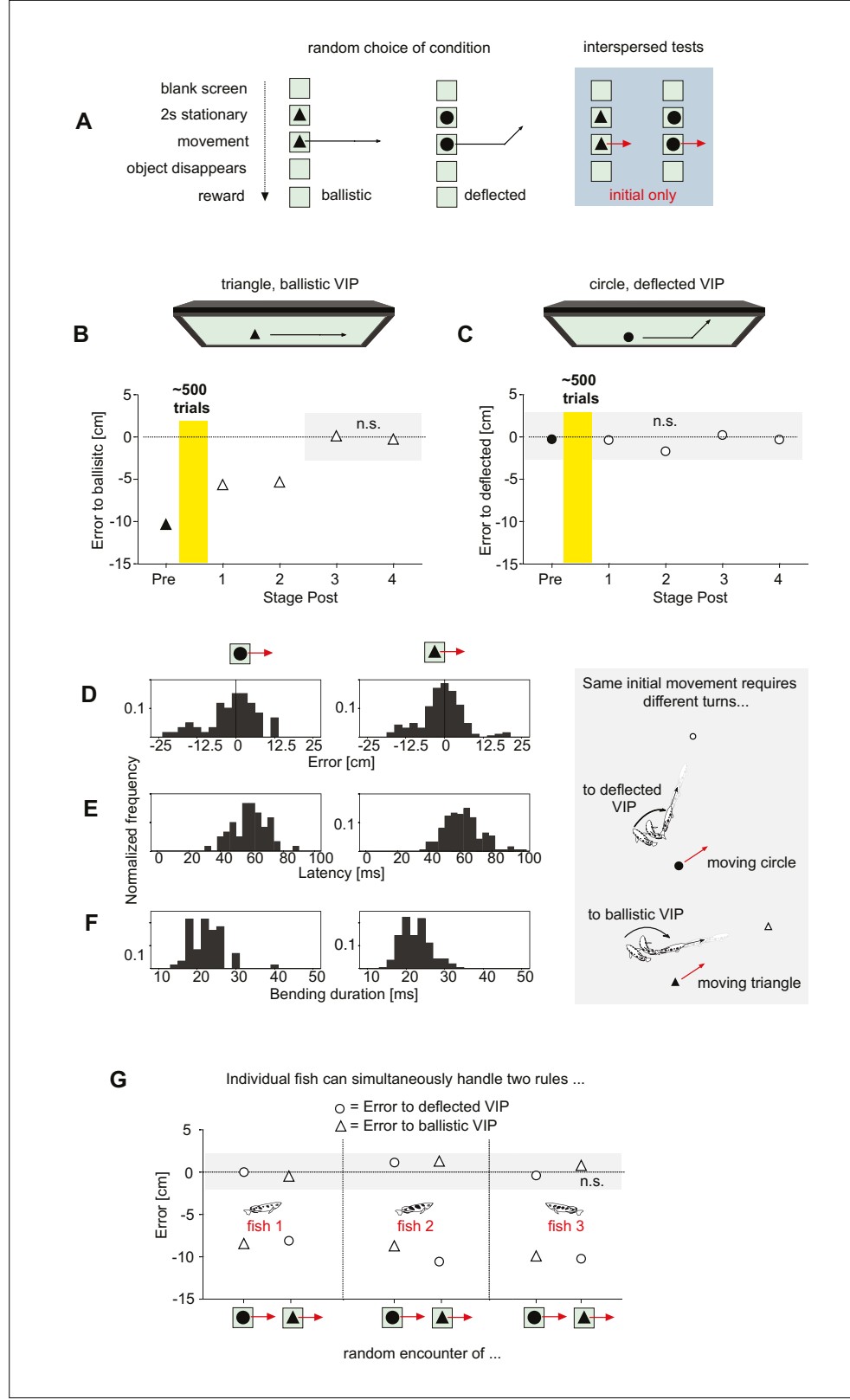

**Figure 6.** The highspeed decision can conditionally use two different rules of how to connect input and output.
(**A**) After successful training to infer the new (deflected) virtual impact points (VIPs), fish were randomly shown
either the disks with the deflected trajectories (reward at deflected VIPs) or new non-disk objects that moved on
straight trajectories and were rewarded at the corresponding ballistic VIPs. (**B, C**) Error in the turns as determined

*Figure 6 continued on next page*

*Figure 6 continued*

in interspersed tests that only showed the straight initial movement both for non-disk objects (triangle, **B** and for the familiar disk objects **C**). Objects were chosen randomly. 'Pre' (filled symbols) denotes baseline before fish were exposed to a total of 500 presentations of the non-disk objects with reward at their ballistic VIPs. Errors were determined in subsequent test phases. As the fish changed their turn decision to minimize the error to the non-disk VIPs (**B**) they continued to aim to the deflected VIPs whenever they encountered moving disks (**C**) showing that they had not reversed to generally using ballistics again. (**D–F**) Interspersed tests that only showed the same initial linear trajectory but with either a disk (n=60) or a triangle moving (n=187). Graphs on right side to illustrate choice situation as seen from above: For the same set of input variables (speed, height, direction) the fish must turn to the ballistic VIP when a triangle is moving, but to the deflected VIP when a circle is moving. Detailed analysis shows that turns were appropriate to the correct VIP with no difference in error (**D**), latency (**E**), or kinematics (bending duration) (**F**). (**G**) Individual fish were able to select the appropriate turn to the deflected VIPs when encountering moving disks and to the ballistic VIP when encountering triangles. Aims were equally appropriate and did not differ in variability. Data are represented as medians (**B, C, G**). n.s. not significant.

The online version of this article includes the following source data for figure 6:

**Source data 1.** Contains all data of *Figure 6*.

encounters of moving triangles the turns minimized the error to the ballistic impact point when a triangle was shown (error to ballistic VIPs p=0.836, error to deflected VIPs p=0.020; One-Sample Signed Rank; *Figure 6B*). However, the turns continued to minimize the error to the deflected impact point whenever the disk was shown (error to ballistic p≤0.001, error to deflected p=0.084, n=187; One-Sample Signed Rank; *Figure 6C*).

Next, we interspersed tests that only showed the initial straight paths both for the disk (n=60) and for the triangle (n=187), i.e., the objects vanished before the deflection would occur for the disks. Picking either the 'ballistic' or the 'deflected' rule to choose the appropriate turn could now only be based on target shape. Again, the turns were toward the ballistic landing point when the triangle was shown (difference from ballistic p=0.075, from deflected p<0.001; One-Sample Signed Rank) and to the deflected landing point when the disk was shown (difference from ballistic p=0.008, from deflected p=0.412; One-Sample Signed Rank) (*Figure 6D*). Also, latency (p=0.189, Mann-Whitney; *Figure 6E*) and kinematics of the turns (bending duration; p=0.984; Mann-Whitney; *Figure 6F*) were the same in the responses to triangles and disks. In other words, the nature of the turn decision was the same, but object shape determined which rule the decision used to determine the appropriate turn given the input data.

In principle these results could have arisen rather simply: some fish could have only responded to disks and others only to triangles, so that no single given individual would have to be able to handle both rules. To check if at least one individual would indeed be able to simultaneously use both rules we analyzed the contributions of the individual fish. Our analysis showed that all three of the three individual fish that had contributed most of the starts in *Figure 6* minimized the error to the appropriate VIP (*Figure 6G*; difference from zero error always p>0.1; One-Sample Signed Rank). Also, the variability of the errors in applying the two rules did also not differ in any of the individuals (p=0.966; Brown-Forsythe). This shows clearly that the decision circuit used by archerfish does have the potential of simultaneously operating on two different rules.

## A shape cue is needed to select which rule to use but need not be available a priori

In the previous series of experiments, the fish could always see the shape of their virtual prey before it started to move. In principle, shape information could, therefore, have acted as a potential prior that would inform the decision circuit which of the two rules to later use when target movement starts. To test whether the circuit relied on such prior information we interspersed tests in which the prey object (disk or triangle) revealed its shape only at motion onset (*Figure 7*). In each of these interspersed tests, first a neutral shape (a cross) appeared, remained stationary for 2 s, and then was exchanged randomly with either the moving triangle or the moving disk. Moreover, only the straight initial path was shown so that shape after motion onset was the only cue available for selecting either the 'ballistic' or the 'deflected' rule (*Videos 6 and 7*). Under these conditions, the decisions remained appropriate to the ballistic (triangle; n=109 starts) or deflected (disk; n=46 starts) VIPs (difference from

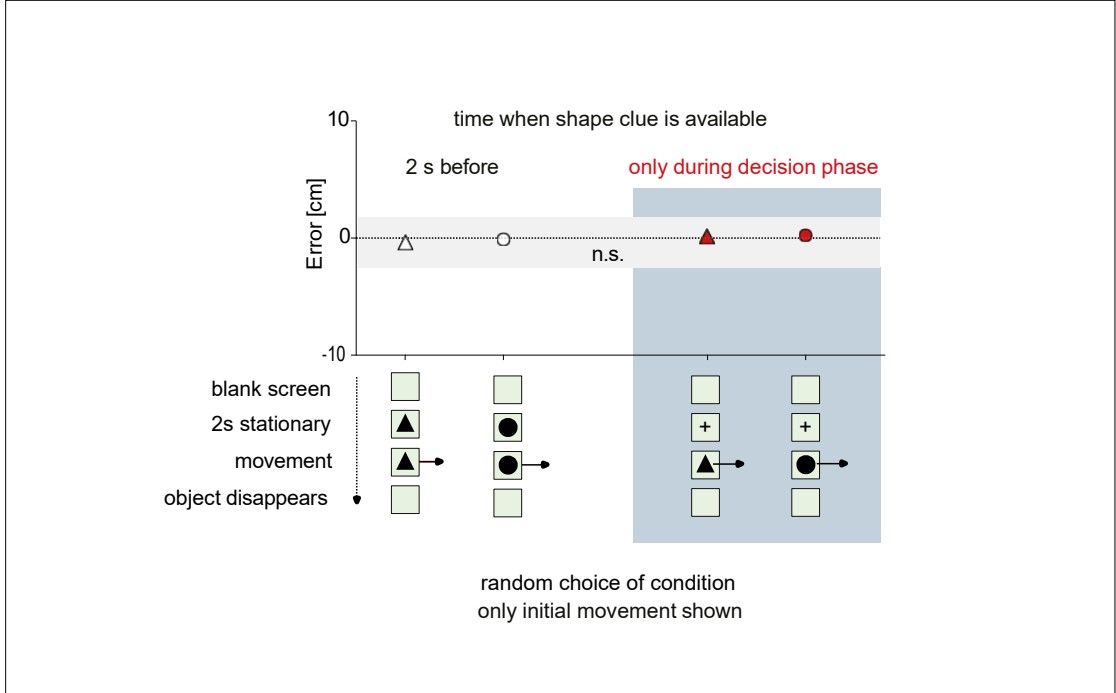

**Figure 7.** The clue that signals which rule must be used is not required before prey motion starts. In this series of experiments, one of two differently shaped types of moving prey are shown, as in *Figure 6*. Either a triangle that required the fish to turn to the appropriate ballistic VIPs, or a disk, that required the fish to turn not according to ballistics but based on the deflected rule (see *Figure 4*). One of the two objects were randomly selected. In one type of experiment ('2 s before'), the decision circuit was given time to adjust to the rule that was needed by displaying the object in stationary position for 2 s prior to motion onset. In the other type ('only during decision phase') no prior information was given. Instead, a neutral symbol (a cross) was shown 2 s before prey motion and then the chosen object (disk or triangle) appeared and immediately was moving. In all tests, only the initial straight movement was shown (i.e. no deflection was visible in the moving disks). Errors were the same, regardless of whether object identity was revealed before or only during the very brief decision time (see *Videos 6 and 7*). Data are represented as medians. n.s. not significant.

The online version of this article includes the following source data for figure 7:

**Source data 1.** Contains all data of *Figure 7*.

zero error to appropriate VIP: p=0.451 (triangle) and p=0.671 (disk); p≤0.001 for the alternatives, all One-Sample Signed Rank). Furthermore, also the distribution of errors was not statistically different from those obtained with the shape cue available 2 s prior to the decision (p=0.269; Brown-Forsythe). Hence the information that is needed to select which rule is to be used can also be extracted in the very brief interval after motion onset, together with all other decision-relevant information. It is not required to instruct the decision circuit before target movement starts.

## Discussion

Using the rapid turn decisions of hunting archerfish as an example we demonstrate that highspeed decisions that demonstrably lack a speed-accuracy tradeoff (*Figure 4J, K*, *Schlegel and Schuster, 2008*), do not necessarily lack flexibility. Using a virtual reality system that exploits key properties of this decision (*Figures 2, 3*) we were able to show that the decision is not hardwired to ballistic falling patterns. Before, the fish would turn to the later ballistic landing point, based on the initial values of prey motion. After a training phase in which the connection between initial motion and the point of reward had been systematically changed – according to a new rule – the fish were able to rapidly turn towards the new points, that were appropriate for their starting position and the combination of initial motion variables. Generalization tests after this reprogramming showed, furthermore, that the decision could immediately handle new constellations never encountered during the re-training phase and that it was capable of simultaneously handling two distinctly different rules, one for each of two visually different prey objects. Our findings provide a striking counterexample to the view that highspeed decision making is generally based on simplifying heuristics and lacks flexibility.

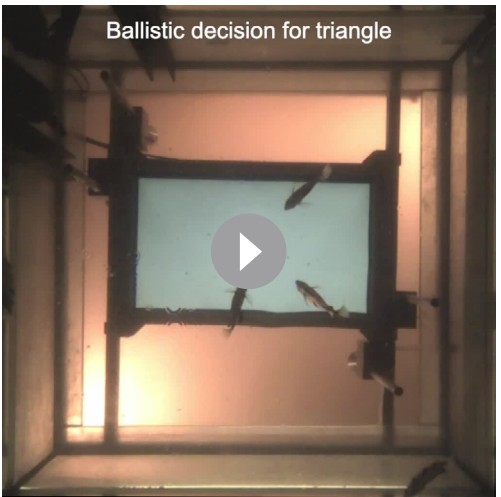

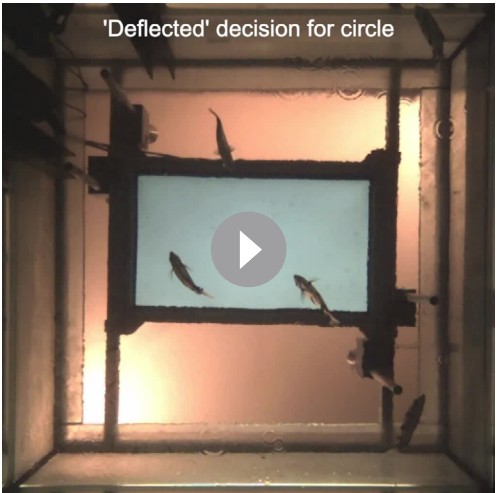

**Video 6.** First example from tests to show the conditional simultaneous use of two different rules. Prior to this recording, all fish had encountered two different types of prey. Each required the use of a different rule to select the appropriate turn to the rewarded impact point, given initial prey motion. In the test shown here, a neutral object (a cross) is displayed on the LCD screen that is located above the tank. After 2 s the cross changes into a triangle (this movie) but it could also have changed into a disk (*Video 7*). In this situation, all information, including that needed to select which rule is to be used, is available only after onset of target movement. The aim at the end of the rapid turn of the first responding fish is indicated by the red line. Regions around ballistic and deflected virtual impact points are highlighted. The scene is filmed from below at 500 fps and shows slowed down 17-fold.

https://elifesciences.org/articles/99634/figures#video6

**Video 7.** Second example from tests to show target-shape dependent conditional use of two rules. Prior to this recording, all fish had encountered two different types of prey. Each required the use of a different rule to select the appropriate turn to the rewarded impact point, given initial prey motion. In the test shown here, a neutral object (a cross) is displayed on the LCD screen that is located above the tank. After 2 s the cross changes into a disk (this movie) but it could also have changed into a triangle (*Video 6*). In this situation, all information, including that needed to select which rule is to be used, is available only after onset of target movement. The aim at the end of the turn of first responding fish is indicated by the red line. Regions around ballistic and deflected virtual impact points are highlighted. The scene was filmed from below at 500 fps and shown slowed down 17-fold. Note, that only the initial straight path of the trajectory is shown in this test, so that the trajectory is completely identical to that of *Video 6* (with a moving triangle).

https://elifesciences.org/articles/99634/figures#video7

## Comparing cognitive aspects of highspeed decision making with those of much slower behaviors

None of the cognitive aspects we describe here is in any way unique and not found in much slower behaviors. For example, archerfish generalize efficiently when they learn how to engage moving targets and to judge the absolute size of prey (*Schuster et al., 2004*; *Schuster et al., 2006*). Also, recognizing and assigning a value to different visual shapes has been demonstrated much earlier (e.g. *Rischawy and Schuster, 2013*) and many aspects have been described across the animal kingdom, including animals with far smaller brains (*Chittka and Niven, 2009*; *Chittka and Jensen, 2011*; *Giurfa, 2013*). What is surprising, therefore, is clearly not to find cognitive features in archerfish, but rather to find such features in a decision that is made at a very different timescale at reflex-like speed (less than 100 ms) and that lacks a speed-accuracy tradeoff (*Figure 4J, K*; *Schlegel and Schuster, 2008*). The complexity involved in reprogramming the archerfish highspeed decision must also not shy a comparison with tasks that are considered highly sophisticated in many laboratory animals (e.g., *Asaad et al., 1998*; *Bunge, 2004*; *Fusi et al., 2007*). For example, a much-studied task that is considered demanding is to have an animal under condition 'a' respond to a stimulus 'α' by turning left and right to stimulus 'β.' Under condition 'b,' however, the animal must do the opposite and turn right to stimulus 'α' and left to stimulus 'β.' In comparison, the task the decision circuit of archerfish can handle in as little as just 40 milliseconds (see *Figure 4*) is not only to conditionally switch between two motor outputs but to switch between two complete rules of how to connect a three-dimensional set of input data to a continuum of possible motor outputs.

## Highspeed decision-making and minibrains

The plausible assumption that reflex speed implies reduced complexity and flexibility (*Fusi et al., 2007*; *Wang, 2008*) could have a similar foundation as the assertion that smaller brains have reduced cognitive capacities. Substantial progress has been made on this latter assertion, resulting in an impressive and continuously growing list of sophisticated cognitive features that were found in animals with bird- and fish brains (*Bshary et al., 2002*; *Shettleworth, 2010*; *Güntürkün and Bugnyar, 2016*). Most impressively, it has been demonstrated that insect brains that operate with a millionth of the neurons of a human brain can handle a wide range of cognitive tasks (*Chittka and Niven, 2009*; *Chittka and Jensen, 2011*; *Giurfa, 2013*; *Giurfa et al., 2001*; *Avarguès-Weber et al., 2012*; *Chittka, 2017*) that even include social or observational learning (*Chittka, 2017*; *Loukola et al., 2017*; *Bridges et al., 2023*). Of particular interest in our context is that it has even been demonstrated that learning can lead to the reprogramming of innate and supposedly hardwired behaviors in insects (*Roussel et al., 2012*).

The reflex speed of some decisions made by animals with much larger brains also will likely limit the size of the underlying neuronal circuitry, allowing only a subset of the brain to be used, a subset that could be considered a 'functional minibrain,' The assertion that everything that is as fast as a reflex must be simple might, therefore, suffer the same fate as the assertion that brains with fewer neurons must be simple. In the turn decisions of archerfish, specifically, it is known that the kinematics of their turn decisions is completely equivalent to the so-called C-starts that the fish produce to escape (*Wöhl and Schuster, 2007*). Both the turn decisions (also called predictive C-starts; *Schuster, 2023*) and the archerfish escape C-starts are among the fastest C-start known in fish (*Wöhl and Schuster, 2007*) and so at least the motor circuits are restricted to the well-known powerful hindbrain circuits of a comparably small number of tractable numbers of neurons, organized around the giant Mauthner neuron (*Korn and Faber, 2005*; *Machnik et al., 2018a*; *Machnik et al., 2018b*; *Hecker et al., 2020*). Its rapidly conducting axon would be suited to initiate the turn but the recruitment of other hindbrain neurons is needed to account for the precision and adjusted straightening speed of the fish (*Wöhl and Schuster, 2007*).

## Are the findings typical for archerfish?

Our aim was to test the idea that the decisions are hardwired. To show that the decision has the potential to be reprogrammed – and that it can even operate on two rules – requires careful experiments and controls, but clearly one fish that can do it illustrates what the circuitry can do in principle. To demonstrate that the physiology of a tiger allows it to be green, observing one such tiger is sufficient. This is why, for example, studies on a single gray parrot e.g., *Pepperberg, 2000* have been extremely valuable. Here, we analyzed a group of six fish that reached homogeneous stages of performances. Clearly, there could be large differences in the learning speed of the individuals. An extremely exciting possibility – that we cannot presently address – is if the reprogramming even would allow for some form of observational or even social learning. This would be highly valuable to speed up the learning and to reprogram the decisions with fewer examples.

We do believe, although the number of individuals that we were able to train and test in the present long endeavor is limited, that our findings are of relevance in the wild. First, rapid and precise turn decisions are required because of the severe competition archerfish often face. Other surface-feeding fish are not only more numerous in Thailand – and, therefore, likely to be closer at the landing point of the archerfish's prey – but they are better equipped with mechanosensors and are also capable of using vision to guide pre-impact turns. It is, therefore, crucial for archerfish to outperform them by using fast and precise turn decisions (*Rischawy et al., 2015*). We suggest that the clearly demonstrated capacity of these decisions for being flexible has evolved because of this competition and because of the scarcity of aerial prey at least in many natural habitats. This requires the fish to down and efficiently secure prey of various sizes. While small prey will typically fall ballistically (*Rossel et al., 2002*; *Reinel and Schuster, 2018a*), larger prey will face considerable frictional forces and – all other conditions being equal – will, therefore, not fall equally far. Simply using hardwired circuitry for ballistically falling prey would produce wrong initial turns for these larger items and would require subsequent corrections that cost precious time and might mean losing some of the advantage over the competitors. The flexibility we found and the ability to operate on at least two types of prey that fall according to two different rules would allow at least some individual archerfish to efficiently handle

two types of prey that co-occur during a specific time of the year. When one is replaced by another type of prey with yet another way in which its initial movement relates to its impact point, the decision would be able to adjust, again, to the new prey type, being still able to accurately respond to the prey that remained. So, we suggest, clearly without proof, that the ecological constraints and the severe competition that archerfish face in the wild could have favored the evolution of a decision that has the potential for plasticity and for combining speed, accuracy, and flexibility.

## Conclusions

Our findings demonstrate that highspeed decisions that lack a speed-accuracy tradeoff are not necessarily limited to operating based on simplifying heuristics but can, at least in principle, offer learning capacities and cognitive aspects of much slower decisions and behaviors that occur at least at 100 times lesser speed. Our findings raise several exciting possibilities. First, it will be exciting to see if the capacities we found in archerfish are special or perhaps much more widespread among other high-speed decisions. Second, if highspeed decisions are not generally incapable of learning and flexibility, the obvious question remains more pressing what might be the other reasons that most decisions do show clear speed-accuracy trade-offs (*Chittka et al., 2009*). Third, the findings suggest that rapid decision-making can be efficiently used in many technological applications in which time is premium – even if such applications require a high degree of flexibility.

## Materials and methods

### Animals

Experiments were conducted on a group of six adult archerfish (*Toxotes chatareus*). The experimental fish were randomly chosen from a larger group of (male and female) archerfish. They had been imported from Thailand (Aquarium Glaser, Rodgau, Germany) and were kept at 26 °C on a 12 hr:12 hr light/dark cycle in the large experimental tank (1 m × 1 m × 0.6 m, filled to a height of 34 cm) in brackish water (3.5 mS cm$^{-1}$). All procedures were conducted in accordance with the German Animal Welfare Act (Tierschutzgesetz) as well as the German Regulation for the protection of animals used for experimental purposes or other scientific purposes (Tierschutz-Versuchstierordnung) and were approved by the government of Lower Franconia (Regierung Unterfranken, Würzburg, Germany: Az. 2531-05-1).

### General setup

To facilitate recording the fish from below – through the transparent bottom of the tank – together with the visual stimuli shown on the LCD monitor above, the tank was placed on steel frame stands, 110 cm above the ground. A construction made from 3 cm aluminum profiles (item Industrietechnik GmbH, Germany) was used to hold the monitor on top of the tank as well as to position the feeders that were employed to reward the fish. To increase the contrast in the recordings of the fish, three conventional 500 W halogen floodlights, located around the tank, illuminated a square of white cloth (2×2 m) that was placed parallel to the water surface and 106 cm above it.

### Monitors and stimuli

To present visual stimuli a 27'' LCD monitor (Samsung SyncMaster S27A750D, 120 fps, max. intensity 300 cd/m$^2$) was used. A glass plate in front of the screen protected it against shots of brackish water fired by the fish. All objects used in this study had a Michelson contrast of 0.96, as determined with a luminance meter (LS-110 Konica Minolta Sensing Europe) under the conditions of the actual experiments. The surface of the LCD monitor was parallel to the water surface, h=28.8 cm above it and its center was directly above the tank's center. Stimuli were generated using Apple Motion 5.4.7 and stored as 120 fps FullHD ProRes movies. We presented the movies with the presentation mode of QQuickTimePlayer 10.4 (Apple Inc) running on a MacPro 5.1 under MacOS 10.11. An AMD 4 GB Radeon RX 480 graphic card produced the monitor signal and was connected via DisplayPort with the LCD monitor. Unless stated otherwise ,each presentation started with an object appearing in the center of the screen. After 2 s ,the object starts moving in one of four pre-assigned, randomly chosen directions. Objects displayed were either disks of 4 mm diameter, disks of 13 mm diameter, or equilateral triangles (height 13 mm). Objects that mimicked ballistically falling motion moved at a

horizontal speed $v_h$ = 1.775 m s$^{-1}$ and their virtual landing occurred 243 ms later, after a (virtual) path of 43 cm. On the screen, the movement of the ballistic objects was visible for 116 ms. Objects on the 'deflected' trajectories moved at the same horizontal speed but changed the direction of motion after being visible for 116 ms (or 20.5 cm) at a (virtual) time of 118.7 ms (i.e. after an initial straight path of $d_r$ = 21.1 cm) by an angle $\gamma$=39.8° to the left (see *Figure 4A*). They were then visible for a further 58 ms. The systematic change in direction causes the virtual impact point, the one inferred from initial motion, to be systematically offset by o=2*($v_h$*sqrt(2 h/g) - $d_r$) * sin($\gamma$/2) where h is the height at motion onset, g is the gravitational constant and $d_r$ is the distance traveled prior to the change in direction. In our experiments, we selected conditions so that the offset was large enough (14.9 cm) to allow us detecti to detectors in the bearing of the C-starts. In critical tests, only the straight initial movement (i.e. the initial path of 20.5 cm) was shown both for ballistic and deflected trajectories.

## Stimulus-coupled feeders

Unless stated otherwise, four feeders were typically present. During training, they were located at the places of possible virtual impact points. In many tests, the virtual impact points were systematically offset from the feeders by changing the direction of motion shown on the screen. This way, the virtual landing points differed both from the position of the feeders as well as from the position relative to landmarks the fish might have been using during training. The feeders were custom-made (*Figure 3—figure supplement 1*) and consisted of a tube (2 cm inner diameter, 44 cm long) with a sliding plate inserted 20 cm below its upper end to block the passage of a food pellet (one half of a Sera Cichlid stick). Each feeder was equipped with an electromagnet that could retract the slider to allow the passage of the pellet. All feeders were pre-loaded at random intervals before the tests, using a funnel (6 cm diameter) on the top side of each feeder (*Figure 3—figure supplement 1*). Which of the four electromagnets was activated at which time was defined by trigger pulses on the channels of the audio track of the movies shown on the LCD monitor so as to achieve an impact of the pellet at the virtual impact point and after the expected time T=sqrt(2 h/g) of falling (h=initial height, g=gravitational acceleration). Four audio channels were used to control which of the four electromagnets was to be activated. To achieve the required number of channels, we connected an AV receiver (Sony STR-DH520) to the computer via HDMI. Four output channels of the AV receiver delivered the trigger signals to a custom-made amplifier which activated the electromagnets by means of a 200 ms pulse (12 V/300 mA). An additional audio channel triggered a pulse generator (TGP110; TTi, England) that started the highspeed video recording.

## Highspeed imaging

All recordings were made from below through the tank's transparent bottom (*Figure 3—figure supplement 1*). A Fastcam MC2 color 500 (Photron, Japan) highspeed camera, operated at 500 fps (512×512 pixels) and equipped with a CVM0411ND f1.6/4.4–11 mm lens (Lensation, Germany) was used to monitor the complete tank. The pre-trigger option of the camera was always used and set to record 4 s before and 4 s after the trigger signal (i.e. the onset of target motion). An Apple MacBook Pro 6.2 running Microsoft Windows 7 controlled the setup via Fastcam Viewer 332. Recordings were stored as QuickTime movies and contained the appearance of the object on the screen, its motion, and the impact of food delivered by one of the feeders.

## Discriminating the responses of individual group members

Because all experiments on the start decisions require the competition within a group (*Schlegel and Schuster, 2008*), we confirmed that all fish in the group always contributed to the decisions reported throughout the study. This was checked at every stage of the work. Details on the individual performance are reported for the last and most demanding task that shows that these individuals had indeed learned everything that we report in this study (*Figure 6*). Individuals were recognized based on the contour of their frontal body as seen from below. A template contour was used for each individual fish that was drawn starting at one pectoral fin, over the snout, and ending at the other pectoral fin. Respective images of the highspeed recordings were extracted using Image J (*Korn and Faber, 2005*) just when the straightening phase (so-called stage 2) of the so-called C-start (*Figure 2—figure supplement 1*) was terminated. Note that the turns are made right below the water surface and thus in a fixed distance from the camera. The contours taken at the end of stage 2 were specific for

each individual fish in the experimental group. Discrimination performed this way coincided with one obtained independently by additionally recording the responses from the side (Photron Fastcam MC2 color 2 K, equipped with same type of lens and also operated at 500 fps, view orthogonal to the front side and 50 cm away from it) in which the individuals could be identified by means of their individual stripe patterns.

### Natural stimuli used to test the virtual reality approach

A non-transparent platform (4.5 cm in diameter) was mounted above the center of the tank. This allowed to blow food pellets off the platform from a similar initial height as the virtual objects shown on the screen. Movement was started by an airflow (as in *Schlegel and Schuster, 2008*; *Reinel and Schuster, 2018a*) and pellets fell ballistically toward the water surface.

### Quantitative analysis of the turn responses

Image J 1.50 (*Schneider et al., 2012*) was used to analyze the recorded videos. To ensure that the decisions could not be influenced by those of other fish, we exclusively analyzed the turn of the fish that responded first in each given test. Latency was the time between onset of movement of the prey object on the LED screen and the onset of the C-start. Error in aim relative to a point of interest (e.g. the virtual impact point) was determined from the last frame at the end of the straightening phase (stage 2) of the C-start, i.e., before the fish starts its approach trajectory (*Figure 2—figure supplement 1*). Orientation of the fish's rigid front end was determined from the tip of the snout and the point midway between the two pectoral fins. If a line drawn along this orientation did cut a second line drawn between the start of motion and the point of interest (e.g. the virtual impact point), both projected to the water surface, then the error was defined as negative, otherwise, it was defined as positive (*Figure 2—figure supplement 1*). Additionally, we determined kinematic aspects of the C-starts, such as the duration of its first bending stage (stage 1 of the C-start), the mean angular speed (i.e. turn angle/turn duration), and the initial orientation and distance the fish had before the onset of their C-start (*Wöhl and Schuster, 2007*). Only C-starts that were completed before prey impact were included in our analysis, to ensure that the fish could not have used information from the impact of the food pellets to select their C-starts. Furthermore, only C-starts were included in which the subsequent approach path of the responding fish was not blocked by another fish. When aims had to be evaluated relative to two points of interest (e.g. virtual landing point and feeder position) then (i) both possible approach paths had to be free and not blocked by any group member and (ii) the responding fish had not to be oriented in a line with both points. Analysis of the coordinates and frame numbers determined with ImageJ was carried out using Excel (14.0 & 15.11).

### Statistics

Statistics were run on GraphPad Prism 9.5. (GraphPad Software, USA). Tests referred to in the main text were as follows: Normality of data was assayed using Shapiro–Wilk tests. For single data sets (error of aim), difference from zero was checked using One-Sample t-tests (parametric) or One-Sample Signed Rank tests (non-parametric). Comparisons of two data sets (latency, duration of stage 1, error of aim) were performed with unpaired t-tests or Mann-Whitney tests. Comparisons of three or more data sets (latency, duration of stage 1, error of aim) were performed with Kruskal–Wallis One-Way ANOVA on ranks, post hoc tested with Dunn's method (non-parametric). Equal variance between data sets (error of aim) was checked using Brown-Forsythe tests (non-parametric). Stability of the aim was tested using linear regression and Spearman's correlation. Tests for changes in C-start probability employed pairwise comparisons with $\chi^2$ tests. All respective tests were performed two-sided. ***p<0.001, **p<0.01, n.s. not significant. All data are available as source data files.

### Acknowledgements

We thank the German Research Foundation (grants Schu 1470/8–1 and Schu 1470/11–1) for continuous support, Peter Machnik and Nick Jones for feedback on the manuscript and our reviewers and editors for their comments.

# Additional information

## Funding

| Funder | Grant reference number | Author |
| --- | --- | --- |
| Deutsche Forschungsgemeinschaft | Schu1470/11-1 | Stefan Schuster |
| Deutsche Forschungsgemeinschaft | Schu1470/8-1 | Stefan Schuster |

The funders had no role in study design, data collection and interpretation, or the decision to submit the work for publication.

## Author contributions

Martin Krause, Data curation, Formal analysis, Investigation, Methodology; Wolfram Schulze, Software, Supervision, Methodology; Stefan Schuster, Conceptualization, Resources, Data curation, Formal analysis, Supervision, Funding acquisition, Visualization, Writing – original draft, Project administration, Writing – review and editing

## Author ORCIDs

Martin Krause ⓘ https://orcid.org/0009-0008-8884-9522
Stefan Schuster ⓘ https://orcid.org/0000-0002-0873-8996

## Ethics

All procedures were conducted in accordance with the German Animal Welfare Act (Tierschutzgesetz) as well as the German Regulation for the protection of animals used for experimental purposes or other scientific purposes (Tierschutz-Versuchstierordnung) and were approved by the government of Lower Franconia (Regierung Unterfranken,Wurzburg, Germany: Az. 2531-05-1).

Reviewer #1 (Public review): https://doi.org/10.7554/eLife.99634.3.sa1
Reviewer #2 (Public review): https://doi.org/10.7554/eLife.99634.3.sa2
Author response https://doi.org/10.7554/eLife.99634.3.sa3

# Additional files

## Supplementary files
MDAR checklist

## Data availability

All data generated or analysed during this study are included in the manuscript and supporting files. Additionally we have uploaded all source data as source data files.

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
