## [Editor Report · eLife assessment]

This **important** study investigates the adaptability of prey capture by archerfish, which hunt insects by spitting at them and then rapidly turning to reach their landing point on the water surface. The results of elaborate behavioral experiments and measurements show that, even though the visuomotor behavior unfolds very rapidly (in less than 100 ms), it is not hardwired and can adapt to different simulated physics and different prey shapes. The data are **convincing** and should be of relevance to those interested in rapid decision making in general, beyond the archerfish model.

---

## [Referee Report · Reviewer #1 (Public review)]

Summary:

The authors test whether the archerfish can modulate the fast response to a falling target. By manipulating the trajectory of the target, they claim that the fish can modulate the fast response. While it is clear from the result that the fish can modulate the fast response, the experimental support for argument that the fish can do it for a reflex like behavior is inadequate.

Strengths:

Overall, the question that the authors raised in the manuscript is interesting.

Weaknesses:

Major comments:

(1) The argument that the fish can modulate reflex-like behavior relies on the claim that the archerfish makes the decision in 40 ms. There is little support for the 40 ms reaction time. The reaction time for the same behavior in Schlegel 2008, is 60-70 ms and in Tsvilling 2012 about 75 ms, if we take the half height of the maximum as estimated reaction time in both cases. If we take the peak (or average) of the distribution as an estimation of reaction time, the reaction time is even longer. This number is critical for the analysis the authors perform since if the reaction time is longer, maybe this is not a reflex as claimed. In addition, mentioning the 40 ms in the abstract is overselling the result. The title is also not supported by the results.

(2) A critical technical issue of the stimulus delivery is not clear. The frame rate is 120 FPS and the target horizontal speed can be up to 1.775 m/s. This produces target jumping on the screen 15 mm each frame. This is not a continuous motion. Thus, the similarity between the natural system where the target experience ballistic trajectory and the experiment here is not clear. Ideally, another type of stimulus delivery system is needed for a project of this kind that requires fast moving targets (e.g. Reiser, J. Neurosci.Meth. 2008). In addition, the screen is rectangular and not circular, so in some directions the target vanishes earlier than others. It must produce a bias in the fish response but there is no analysis of this type.

(3) The results here rely on the ability to measure the error of response in the case of virtual experiment. It is not clear how this is done since the virtual target does not fall. How do authors validate that the fish indeed perceives the virtual target as falling target? Since the deflection is at a later stage of the virtual trajectory, it is not clear what is the actual physics that governs the world of the experiment. Overall, the experimental setup is not well designed.

Comments on revisions:

The authors handled the comments, and the manuscript has improved accordingly. While some issues could still benefit from further clarification and depth, the current version meets the necessary standards.

---

## [Referee Report · Reviewer #2 (Public review)]

Summary:

This manuscript studies the prey capture by archer fish, which observe the initial values of motion of aerial prey they made fall by spitting on them, and then rapidly turn to reach the ballistic landing point on the water surface. The question raised by the article is whether this incredibly fast decision-making process is hardwired and thus unmodifiable or can be adjusted by experience to follow a new rule, namely that the landing point is deflected from a certain amount from the expected ballistic landing point. The results show that the fish learn the new rule and use it afterwards in a variety of novel situations that include height, side and speed of the prey, and which preserve the speed of the fish's decision. Moreover, a remarkable finding presented in this work is the fact that fish that have learned to use the new rule can relearn to use the ballistic landing point for an object based on its shape (a triangle) while keeping simultaneously the 'deflected rule' for an object differing in shape (a disc); in other words, fish can master simultaneously two decision-making rules based on the different shape of objects.

Strengths:

The manuscript relies on a sophisticated and clever experimental design that allows changing the apparent landing point of a virtual prey using a virtual reality system. Several robust controls are provided to demonstrate the reliability and usefulness of the experimental setup.

Overall, I like very much the idea conveyed by the authors that even stimuli triggering apparently hardwired responses can be relearned in order to be associated to a different response, thus showing the impressive flexibility of circuits that are sometimes considered as mediating pure reflexive responses. This is the case - as an additional example - of the main component of the Nasanov pheromone of bees (geraniol), which triggers immediate reflexive attraction and appetitive responses, and which can, nevertheless, be learned by bees in association with an electric shock so that bees end up exhibiting avoidance and the aversive response of sting extension to this odorant(1), which is a fully unnatural situation, and which shows that associative aversive learning is strong enough to override preprogrammed responding, thus reflecting an impressive behavioral flexibility.

Weaknesses:

As a general remark, there is some information that I missed and that are mandatory in the analysis of behavioral changes: one is the variability in the performances displayed. The authors mentioned that the results reported come from 6 fish (which is a low sample size). How were the individual performances in terms of consistency? Were all fish equally good in adjusting/learning the new rule? How did errors vary according to individual identity? It seems to me that this kind of information should be available as the authors reported that individual fish could be recognized and tracked (see lines 620-635) and is essential for appreciating the flexibility of the system under study.

The other information that I could not find explained in a proper way is referred to the speed of the learning process. Admittedly, fish learn in an impressive way the new rule and even two rules simultaneously; yet, how long did they need to achieve this? In the article, Fig 2 mention that at least 6 training stages (each defined as a block of 60 evaluated turn decisions, which actually shows that the standard term 'Training Block' would be more appropriate) were required for the fish to learn the 'deflected rule'. While this means 360 trials (turning starts), I was left with the question of how long did this process last? How many hours, days, weeks were needed for the fish to learn? And as mentioned above, were al fish equally fast in learning? I would appreciate explaining this very important point because learning dynamics is relevant to understanding the flexibility of the system.

Comments After Revision:

There was consensus among reviewers that the authors addressed the initial critiques adequately and that the manuscript improved accordingly. The revision clarified several methodological aspects, and the addition of the new Fig. 2 was particularly helpful. It elucidates the experimental approach used in the study and offers essential context for understanding points that may have been unclear in the previous version.

---

## [Author Response]

The following is the authors’ response to the original reviews

**eLife Assessment**
This valuable study investigates prey capture by archer fish, showing that even though the visuomotor behavior unfolds very rapidly (within 40-70 ms), it is not hardwired; it can adapt to different simulated physics and different prey shapes. Although there was agreement that the model system, experimental design, and main hypothesis are certainly interesting, opinions were divided on whether the evidence supporting the central claims is incomplete. A more rigorous definition and assessment of "reflex speed", more detailed evidence of stimulus control, and a more detailed analysis of individual subjects could potentially increase confidence in the main conclusions.

Thank you very much. There are several points that we had to absolutely make sure that they are very well understood. (1) Explaining in the best possible way the experiment with a fly sliding on top of a glass plate. Here, the virtual ballistic landing point can be calculated using simple high school physics. It turns out that this is where the fish turn to – even though the fly is not falling at all. Once this is understood it becomes clear that we can precisely measure latency and accuracy of the C-start turns. In the new version we explain this essential aspect in more detail and add an extra Figure (new Figure 2). This may, perhaps, help readers to notice this important background (previously covered in Fig. 1C). (2) The full experimental evidence that the VR method works is presented in more detail and all measurements necessary will be clear after the new Figure 2. They will however not be clear if this Figure is ignored. (3) We have rewritten the manuscript to make it easier to understand what we wanted to show, why we needed VR to proceed and why the archerfish highspeed decision lent itself so readily to tackle the problem. (4) We emphasize the importance of speed-accuracy tradeoffs in standard decision-making and also include data on the absence of such a relation in the archerfish highspeed decisions.

So, in summary, we have emphasized what we wanted to show and what we did not want to show, we have rewritten the text to make it easier for future readers and we have tried to add more guidance to the figures. We do hope very much that the beauty of the quite unexpected findings is more easily visible to those who take the trouble of actually reading the paper.

**Public Reviews:**

**Reviewer #1 (Public review):**
Summary:The authors test whether the archerfish can modulate the fast response to a falling target. By manipulating the trajectory of the target, they claim that the fish can modulate the fast response. While it is clear from the result that the fish can modulate the fast response, the experimental support for the argument that the fish can do it for a reflex-like behavior is inadequate.

Please note that we have not simply tested whether archerfish can 'modulate the fast response'. We quantitatively test specific hypotheses on the rules used by the fish. For this the accuracy of the decisions is analyzed with respect to specific points that can be calculated precisely in each of the experiments. These points are shown on the figures and in the movies that were meant to illustrate this important aspect. We had to make sure that the way we calculate the predicted point(s) is made as clear as possible in the text. We added more text and separated the fundamentally important aspects in a separate Figure 2 to make it more difficult to overlook the fundamental aspects that lay the foundation for everything that follows.

Strengths:Overall, the question that the authors raised in the manuscript is interesting.

Thank you and we do hope very much that, with our revision, you will see the beauty of the findings.

Weaknesses:(1) The argument that the fish can modulate reflex-like behavior relies on the claim that the archerfish makes the decision in 40 ms. There is little support for the 40 ms reaction time. The reaction time for the same behavior in Schlegel 2008, is 6070 ms, and in Tsvilling 2012 about 75 ms, if we take the half height of the maximum as the estimated reaction time in both cases. If we take the peak (or average) of the distribution as an estimation of reaction time, the reaction time is even longer. This number is critical for the analysis the authors perform since if the reaction time is longer, maybe this is not a reflex as claimed. In addition, mentioning the 40 ms in the abstract is overselling the result. The title is also not supported by the results.

Although the minimum latency is indeed 40 ms (it can be slightly less: e.g., see the evidence in the paper, for instance the plots in the new Fig. 4) the paper's statements are not dependent on a specific number. Even if minimum latency was 100 ms (which it is not) the speed of the response and the absence of a speedaccuracy relation (now shown directly in Fig. 4) is what is of importance. To show this we have completely rewritten large parts of the manuscript.

(2) A critical technical issue of the stimulus delivery is not clear. The frame rate is 120 FPS and the target horizontal speed can be up to 1.775 m/s. This produces a target jumping on the screen 15 mm in each frame. This is not a continuous motion. Thus, the similarity between the natural system where the target experiences ballistic trajectory and the experiment here is not clear. Ideally, another type of stimulus delivery system is needed for a project of this kind that requires fast-moving targets (e.g. Reiser, J. Neurosci.Meth. 2008). In addition, the screen is rectangular and not circular, so in some directions, the target vanishes earlier than others. It must produce a bias in the fish response but there is no analysis of this type.

Please note that the new Fig. 3 (former Fig. 2) reports all the evidence that is needed to just show this and in a way that could in no way have been better. We have rewritten the text to explain what needs to be shown experimentally in order to be able to proceed, what critical tests were done and what results were obtained. We also add a short comment on another unsuccessful attempt that we have tried before.

(3) The results here rely on the ability to measure the error of response in the case of a virtual experiment. It is not clear how this is done since the virtual target does not fall. How do the authors validate that the fish indeed perceives the virtual target as the falling target? Since the deflection is at a later stage of the virtual trajectory, it is not clear what is the actual physics that governs the world of the experiment. Overall, the experimental setup is not well designed.

Understanding this aspect is essential. If the glass plate experiment is not thoroughly understood (new Fig. 2 with new text to emphasize that this is absolutely essential) nothing that follows makes any sense, including what is meant by the statement that the decision could be hardwired to ballistic motion.

**Reviewer #2 (Public review):**
Summary:This manuscript studies prey capture by archer fish, which observe the initial values of motion of aerial prey they made fall by spitting on them, and then rapidly turn to reach the ballistic landing point on the water surface. The question raised by the article is whether this incredibly fast decision-making process is hardwired and thus unmodifiable or can be adjusted by experience to follow a new rule, namely that the landing point is deflected from a certain amount of the expected ballistic landing point. The results show that the fish learn the new rule and use it afterward in a variety of novel situations that include height, side, and speed of the prey, and which preserve the speed of the fish's decision. Moreover, a remarkable finding presented in this work is the fact that fish that have learned to use the new rule can relearn to use the ballistic landing point for an object based on its shape (a triangle) while keeping simultaneously the 'deflected rule' for an object differing in shape (a disc); in other words, fish can master simultaneously two decisionmaking rules based on the different shape of objects.Strengths:The manuscript relies on a sophisticated and clever experimental design that allows changing the apparent landing point of a virtual prey using a virtual reality system. Several robust controls are provided to demonstrate the reliability and usefulness of the experimental setup.Overall, I very much like the idea conveyed by the authors that even stimuli triggering apparently hardwired responses can be relearned in order to be associated with a different response, thus showing the impressive flexibility of circuits that are sometimes considered mediating pure reflexive responses.

Thank you so much for this precise assessment of what we have shown!

This is the case - as an additional example - of the main component of the Nasanov pheromone of bees (geraniol), which triggers immediate reflexive attraction and appetitive responses, and which can, nevertheless, be learned by bees in association with an electric shock so that bees end up exhibiting avoidance and the aversive response of sting extension to this odorant (1), which is a fully unnatural situation, and which shows that associative aversive learning is strong enough to override preprogrammed responding, thus reflecting an impressive behavioral flexibility.

That's very interesting, thanks and we are very happy to mention this important study in the revised version.

Weaknesses:As a general remark, there is some information that I missed and that is mandatory in the analysis of behavioral changes.Firstly, the variability in the performances displayed. The authors mentioned that the results reported come from 6 fish (which is a low sample size). How were the individual performances in terms of consistency? Were all fish equally good in adjusting/learning the new rule? How did errors vary according to individual identity? It seems to me that this kind of information should be available as the authors reported that individual fish could be recognized and tracked (see lines 620-635) and is essential for appreciating the flexibility of the system under study.Secondly, the speed of the learning process is not properly explained. Admittedly, fish learn in an impressive way the new rule and even two rules simultaneously; yet, how long did they need to achieve this? In the article, Figure 2 mentions that at least 6 training stages (each defined as a block of 60 evaluated turn decisions, which actually shows that the standard term 'Training Block' would be more appropriate) were required for the fish to learn the 'deflected rule'. While this means 360 trials (turning starts), I was left with the question of how long this process lasted. How many hours, days, and weeks were needed for the fish to learn? And as mentioned above, were all fish equally fast in learning? I would appreciate explaining this very important point because learning dynamics is relevant to understanding the flexibility of the system.

First, it is very important to keep the question in mind that we wanted to clarify: Does the system have the potential to re-tune the decisions to other non-ballistic relations between the input variables and the output? This would have been established if one fish was found capable of doing that. We have rewritten the introduction and discussion to specifically say what our aim was. We feel that the paper is already extremely long and difficult to understand (even after we tried very hard in this revision to explain everything in detail and as good as we could), requires the establishment of a method whose success was really unexpected and finding a degree of plasticity that we did not expect at all. We also have added a section in the discussion stating what we can, and we cannot say given the number of fish examined. For instance, we do not know if there are differences in the speed at which the different individuals mastered the new rules and if social learning could play a role to speed up the acquisition. That is a brilliant idea and we are very interested in checking this - but we wanted to stick with the (quite ambitious) goal of the present study.

Reference:(1) Roussel, E., Padie, S. & Giurfa, M. Aversive learning overcomes appetitive innate responding in honeybees. Anim Cogn 15, 135-141, doi:10.1007/s10071011-0426-1 (2012).

Thanks for this reference!

**Recommendations for the authors:**

**Reviewer #1 (Recommendations for the authors):**
Minor comments:(1) What is the difference between Reinel, J. Exp. Bio. 2016 and the current study?

Clearly in that study all objects were strictly falling ballistically, and latency and accuracy of the turn decisions were determined when the initial motion was not only horizontal but had an additional vertical component of speed. The question of that study was if the need to account to an additional variable (vertical speed) in the decision would affect its latency or accuracy. The study showed that also then archerfish rapidly turn to the later impact point. It also showed that accuracy and latency were not changed by the added degree of freedom.

(2) How do Figures 2 F and G demonstrate that an accurate start is possible?

See above.

(3) Figure 4 is hard to follow, it is not clear what is presented and how it supports the claim that the new rule is represented in a way that allows immediate generalization.

Yes, this is not at all an easy experiment. Briefly, fish were re-trained at only one height level and then are tested at other levels. The strategy is as in the experiments Schuster et al. 2004 Current Biology, Vol. 14, 1565–1568, Figure 5. We have changed text and Figure (new Figure 5) to show how the predictions were reached.

**Reviewer #2 (Recommendations for the authors):**
Minor remarksLines 88-90: I was surprised to see that in this section, the authors did not mention the speed-accuracy trade-off off which has inspired numerous experiments in animal behavior (1). This could be used to back their point, namely, that speed comes with an apparent cost of a loss in accuracy.

Yes, that is a crucial aspect that was completely missing even though it demonstrates a key aspect of 'standard' versus some 'highspeed' decisions! We definitely had to include it and also to show, directly under the conditions of our present experiments (in the new Fig. 4) the absence of a significant speedaccuracy relation for the archerfish highspeed decisions! Thank you very much for emphasizing this crucial aspect!

Lines 182-184: Specify that this situation corresponds to the hatched bar in Figure (this can be specified in the caption of the figure, where the bar is not mentioned).

Thanks!

Lines 187-188: here and elsewhere (e.g. lines 224-225, etc), the error made by the fish is presented in cm (see Figure 2 where the inset shows how the error was computed). I wonder if it would not be more appropriate to present it in terms of the angular difference between the trajectory made by the fish and the food delivery location.

Angles could also be used, but because of the large variation in initial distances (that we wanted to make sure that the fish had to capture a rule, allowing them to respond from various distances) another measure was used that we found somehow more natural: it is simply how close a fish would get to the landing point if it continued in the direction assumed after the turn. Although we describe how we defined accuracy we did not discuss why this measure was used in this (and many previous studies). We are very happy to add this. Please also note that running all tests based on angular errors (which we also have done throughout to ensure that the conclusions are independent on an arbitrary measure of the error) leads to no different conclusion. We have added a brief explanation in the text and in the new Fig. 2.

Lines 299-323: Is it my impression or did fish have more trouble in generalizing their learned rule to the condition untrained larger height (see for instance red curves in Figures 4 D, E, G)? Could the authors elaborate on this point?

We changed the code to make this more clear. The red curves (before marked A to highlight impact point option A) correspond to the errors to the ballistic impact point without deflection, so what would have to be compared are the black curves (marked P to highlight the virtual impact point that should be chosen had the fish immediately generated to the untrained conditions). We have rewritten the text and the labels in the Figure (now Figure 5) to illustrate the predictions and to name them in more helpful ways and so that they can't be confused with panel labels. At any rate, what needs to be compared, to check the idea, are the black curves, and these are not statistically different between both heights (p=0.525, Mann-Whitney). Interestingly, none of the black curves from all panels (D-G) differ (p>0.3).

Line 559: if we are speaking here about luminance contrast, it should read 'Michelson Contrast' rather than 'Michelsen Contrast'.

Absolutely, thanks!

References(1) Chittka, L., Skorupski, P. & Raine, N. E. Speed-accuracy tradeoffs in animal decision making. Trends Ecol Evol 24, 400-407, doi:10.1016/j.tree.2009.02.010 (2009).

An excellent paper that helps to stress our main question